# Experimentally validating sabatier plot by molecular level microenvironment customization for oxygen electroreduction

Bingyu Huang[1,2], Qiao Gu[1], Xiannong Tang[1], Dirk Lützenkirchen-Hecht[3], Kai Yuan [1] ✉ & Yiwang Chen [1,2] ✉

Microenvironmental modifications on metal sites are crucial to tune oxygen reduction catalytic behavior and decrypt intrinsic mechanism, whereas the stochastic properties of traditional pyrolyzed single-atom catalysts induce vague recognition on structure-reactivity relations. Herein, we report a theoretical descriptor relying on binding energies of oxygen adsorbates and directly associating the derived Sabatier volcano plot with calculated overpotential to forecast catalytic efficiency of cobalt porphyrin. This Sabatier volcano plot instructs that electron-withdrawing substituents mitigate the over-strong *OH intermediate adsorption by virtue of the decreased proportion of electrons in bonding orbital. To experimentally validate this speculation, we implement a secondary sphere microenvironment customization strategy on cobalt porphyrin-based polymer nanocomposite analogs. Systematic X-ray spectroscopic and in situ electrochemical characterizations capture the pronounced accessible active site density and the fast interfacial/outward charge migration kinetics contributions for the optimal carboxyl group-substituted catalyst. This work offers ample strategies for designing single-atom catalysts with well-managed microenvironment under the guidance of Sabatier volcano map.

Aqueous Zn-air batteries (ZABs) are considering an appealing and cost-effective electrochemical energy storage system. Despite ZABs have been intensively developed, the sluggish kinetics of cathodic oxygen reduction reaction (ORR) still demands expensive and complicated catalyst systems promote, and the limited reserve issue of commercial platinum-based ORR catalyst motivates the exploitation of Pt-free materials[1–5]. Atomically dispersed transition metal-nitrogen-carbon (M-N-C) catalysts are viable candidates with promising reactivity, since M-N-C moieties provide unoccupied orbitals that can easily accommodate electron donation from oxygen intermediates[6,7]. While seeking Pt-like ORR intrinsic reactivities, the rational customization of the local physicochemical microenvironment on the central metal is essential to modulate adsorption interaction with oxygen intermediates for M-N-C catalysts[8,9]. Crucially, the structural homogeneity of M-N-C catalysts permits accurate identification and characterization of catalytic sites, which is beneficial for deepening the comprehension of the nature of electrocatalysis[10,11]. Hitherto, M-N-C materials are typically fabricated through pyrolysis treatments on metal-containing precursors, thus posing grand challenges in synthetic control and retaining predesigned geometry[12]. The morphological deformation and even destruction during pyrolysis bring dilemma in realizing structure maintenance and efficient utilization of catalytic metal sites. Moreover, precisely and manageably imbedding specified dopants into appointed positions within the carbon matrix is still a formidable task for

---

[1]College of Chemistry and Chemical Engineering/Film Energy Chemistry for Jiangxi Provincial Key Laboratory (FEC), Nanchang University, Nanchang 330031, PR China. [2]College of Chemistry and Materials/Key Lab of Fluorine and Silicon for Energy Materials and Chemistry of Ministry of Education, Jiangxi Normal University, Nanchang 330022, China. [3]Faculty of Mathematics and Natural Sciences-Physics Department, Bergische Universität Wuppertal, Gauss-Str. 20, D-42119 Wuppertal, Germany. ✉e-mail: kai.yuan@ncu.edu.cn; ywchen@ncu.edu.cn

carbonaceous materials, hence hinders the rational design and syntheses of efficient ORR electrocatalysts. Thereby, propelling the investigation on accurate structure-property-activity relations establishment using well-defined M-N-C moieties as model systems is desirable[12,13].

To improve the ORR efficiency, transition metal porphyrin- and phthalocyanine-based electrocatalysts are widely explored and classified as prospective model systems to achieve atom-level control of molecule moieties with well-manipulated electronic properties and spatial configuration arrangements[14–16]. Thereinto, porphyrin enables systematical modifications at meso-, β-positions and the secondary coordination sphere, while persisting original geometries and coordination characteristics nearly invariable[17–19]. These features afford porphyrin a tailor-made platform to identify the electronic and configurational dynamic cycles on the central catalytic metal, thus demystifying the relevant structure-property correlation and underlying reaction mechanism.

As for representative metal porphyrin-based catalysts, whether dioxygen undergoes fluent chemisorption and activation on an active center is well related to the hybridization status between localized metal 3$d$ electrons and oxygen itinerant 2$p$ electrons. Therein, the chemical microenvironment character of the metal center plays the most prominent part in determining the ORR catalytic efficacy, and cobalt porphyrin turns out to be the most satisfied porphyrin moiety due to its adaptable orbital matching with $O_2$[20–24]. Yet, Co porphyrin-based catalysts still suffer from a high overpotential ($\eta$) in order to exhibit excellent turnover frequency (TOF). In view of this, considerable strategies have put forward to ameliorate their reactivity, which mainly relies on the structural diversity of porphyrin to trigger electronic structural changes on the Co center, such as assembling heteroatom dopants[25], introducing functional substituents[23,26–28], expanding geometry[29] and axial coordination[30–32]. In essence, the electronic arrangement of hybridized orbitals should be customized to establish an appropriate binding interaction between the metal center and diverse oxygen adsorbates for lowering the kinetic energy barrier. More broadly, since the associative ORR pathway involves *OOH, *O, and *OH intermediates, the catalytic theoretical $\eta$ ought to be a function of three microenvironment-dependent adsorption energies[33–36]. Consequently, the linear correlations between adsorption free energies of different oxygen intermediates help to comprehend the ORR activity trend, moreover, the derived Sabatier-type volcano plots are of great significance for forecasting intrinsic performance and simplifying the exploration for advanced catalysts[33,37–39]. However, the Sabatier volcano map has rarely been validated due to the experimentally poor attainability of clear-structured M-N-C materials. Hence, integrating the descriptor-based analysis of absorbates binding with the merits of controllable synthesis and easy-regulated microenvironment, porphyrin system demonstrates great potential to exploit structure-property-activity relations for guiding optimum electrocatalysts.

In this work, we theoretically present a descriptor based on the correlation between the ORR $\eta$ and adsorption behaviors of oxygen intermediates (*OOH, *O, *OH) on the Co-N$_4$ center of Co porphyrin model, and the descriptor-derived Sabatier-type volcano map enables convenient reactivity prediction. The theoretical calculation proposes that the incorporation of electron-withdrawing substituents in the porphyrin secondary coordination sphere is deemed to alleviate the overmuch binding strength of *OH species due to the decreased proportion of electrons in bonding orbital. Therefore, the carboxyl-substituted porphyrin model with an electron-deficient Co center is expected to approach the top of the Sabatier volcano map and emerge with the highest ORR reactivity. To experimentally validate this Sabatier plot, Co porphyrin-based polymer nanocomposite analogs (CoCOP-X@KB, X = CH$_3$, H, COCH$_3$, COOCH$_3$, COOH, CN) were artfully prepared via secondary sphere microenvironment customization. For electrocatalytic ORR, the optimal CoCOP-COOH@KB

shows appealing half-wave potential of 0.86 V and mass activity of 54.9 A g$^{-1}$ @0.8 V. The assembled CoCOP-COOH@KB-based ZAB delivers large peak power density (239.6 mW cm$^{-2}$), high specific capacity (807.1 mAh g$_{Zn}^{-1}$) and long cycling lifespan of over 300 hours. Systematic X-ray spectroscopic and in situ electrochemical characterizations attribute its intrinsic TOF (4.6 e s$^{-1}$ site$^{-1}$) to the considerable accessible active site density (SD; 7.3 × 10$^{19}$ sites g$^{-1}$), as well as the rapid interfacial charge transfer and outward migration kinetics. In situ electrochemical spectroscopic techniques further unfold oxygen intermediates adsorption behaviors and dynamic evolution on the Co-N$_4$ center, also affirm reductive OH* release as the potential determining step.

## Results

### Theoretical study on microenvironment customization

The scaling correlations among Gibbs free energies of oxygen intermediates ($\Delta G_{*OOH}$, $\Delta G_{*O}$, $\Delta G_{*OH}$) may serve as descriptors to facilitate high-efficiency ORR catalysts screening. Primarily, $\Delta G_{*OOH}$, $\Delta G_{*O}$ and $\Delta G_{*OH}$ of Co porphyrin models were simulated to construct a correlation with the overpotential ($\eta$) factor (Fig. 1a, b). The adsorption energies of multiple ORR intermediates are strongly related and difficult to decouple due to the scaling relations[37,40]. The slope of linear correlation between $\Delta G_{*OOH}$ and $\Delta G_{*OH}$ is close to unity since the electron transfer between both *OH, *OOH and Co center is nearly 1 electron, thus the $\eta$ can be determined by $\Delta G_{*OH}$ and ($\Delta G_{*O}-\Delta G_{*OH}$) values (Supplementary Figs. 1–3). As uncovered in Fig. 1a, the color-filled contour map is divided into disparate regions, which are commanded by different potential determining steps (PDSs), containing the reactions $O_2 \rightarrow$ *OOH, *OOH $\rightarrow$ *O, and *OH $\rightarrow$ OH$^-$, respectively. For most atomically dispersed cobalt-nitrogen-carbon (Co-N-C) catalysts, the PDS could be assigned to the *OH release step (*OH $\rightarrow$ OH$^-$), which is intimately associated with the nature of Co orbital configuration. Apparently, the pristine Co porphyrin model (Por-H) locates in the *OH $\rightarrow$ OH$^-$ region, resulting in a high coverage by oxygenated intermediates and difficult subsequent OH$^-$ desorption on the Co-N$_4$ center. As a result, the overmuch oxygen binding limits the ORR kinetics on Por-H model with a large theoretical $\eta$ of 0.44 V. Hence the crux to optimize the electrochemical performance of individual Co-N$_4$ centers is to ameliorate the strong adsorption of oxygen intermediates.

To modulate the electronic microenvironment and corresponding oxygen adsorbate stabilization on Co-N$_4$ center, Co porphyrin models containing five substituents with varied electron-withdrawing/donating properties on secondary coordination sphere were systematically built (Por-CH$_3$, Por-COCH$_3$, Por-COOCH$_3$, Por-COOH and Por-CN). The ORR performances of substituted porphyrin model analogs emerge distinct Sabatier-type volcano relations, in which the models closer to the volcano top (pink color area) own lower $\eta$ and better reactivity (Fig. 1b). Among all Por-X model analogs (X = CH$_3$, H, COCH$_3$, COOCH$_3$, COOH, CN), electron-withdrawing carboxyl substituent renders Co-N$_4$ center the most appropriate intermediate binding with an ideal calculated $\eta$ value of 0.36 V. It is manifested that the secondary sphere substituent significantly customizes the catalytic Co-N$_4$ site and Por-COOH model is resultantly supposed to be the most promising architecture for efficient Co porphyrin-based ORR electrocatalyst.

The ORR processes on Co-N$_4$ configurations were explored from the free energy along thermodynamic pathways and electronic structure to probe the origin of ORR performance enhancement. According to the theoretical calculation results, all the oxygen-bound intermediates only adsorb onto individual Co atoms, which afford superior catalytic centers for ORR. To affirm the effect of secondary sphere substituents on reaction thermodynamics, the ORR Gibbs free energy diagrams ($U = 0$ V) of six customized porphyrin models were computed (Fig. 1c and Supplementary Figs. 4–9). The smallest energy shift observed in free energy diagrams is labeled as reaction energy barrier of the respective model compound, which is important to appraise the

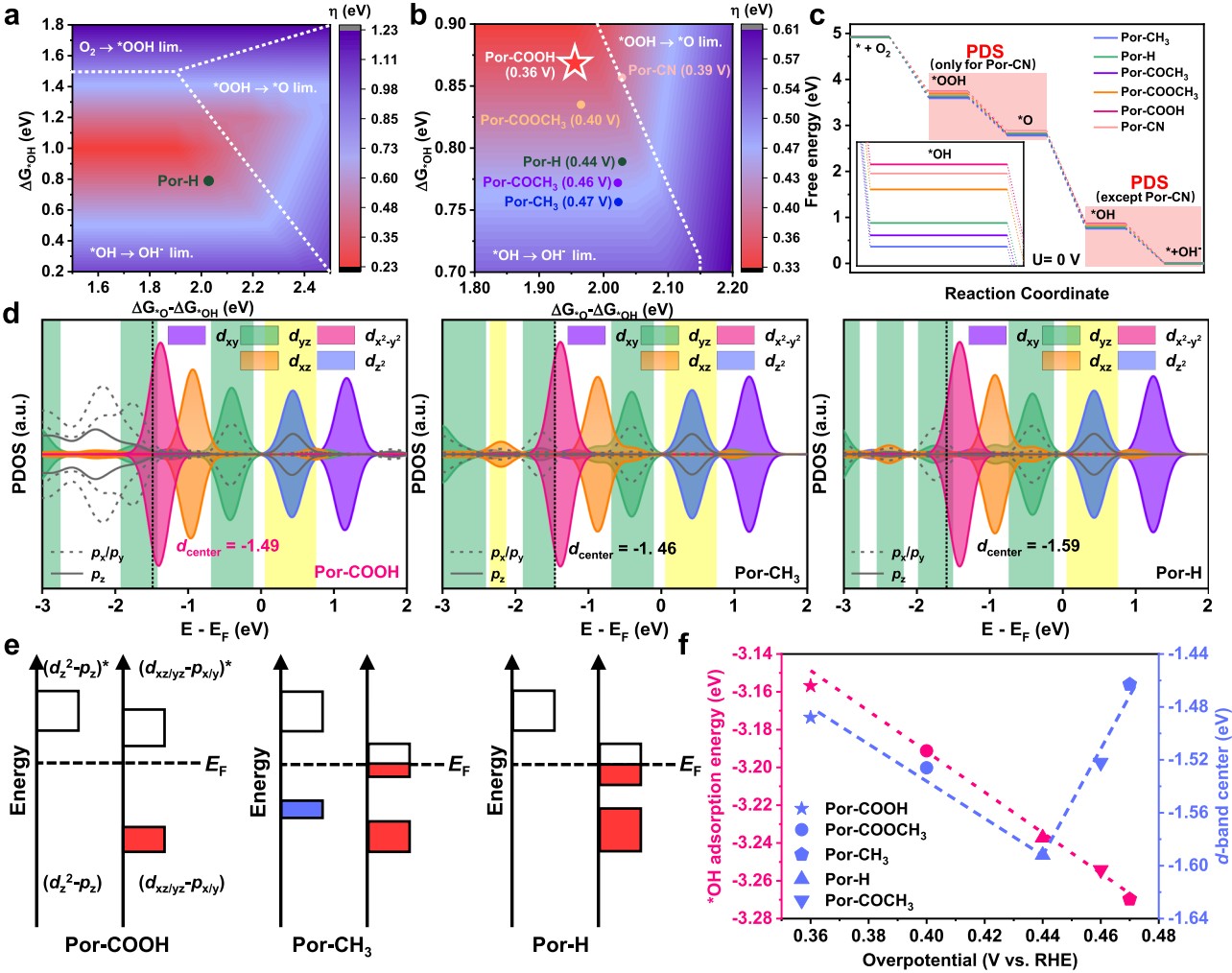

**Fig. 1 | The Sabatier volcano plot and the theoretical predictions of Por-X model analogs.** The color-filled contour maps of theoretical $\eta$ for **a** Por-H and **b** Por-X analogs (X = CH$_3$, H, COCH$_3$, COOCH$_3$, COOH, CN) as a function of $\Delta G_{*O} - \Delta G_{*OH}$ and $\Delta G_{*OH}$. The white dashed lines separate three regions corresponding to different PDSs. **c** The free energy diagram of Por-X analogs at $U = 0$ V. **d** pDOS plots of *OH on Por-COOH, Por-CH$_3$ and Por-H models. Green areas present the overlap between *OH $p_x/p_y$ orbitals and Co $d_{xz}/d_{yz}$ orbitals. Yellow areas present the overlap between *OH $p_z$ orbital and Co $d_{z^2}$ orbital. **e** Schematic diagram of orbitals interaction between Co 3$d$ ($d_{z^2}$ or $d_{xz/yz}$) and O 2$p$ ($p_z$ or $p_{x/y}$) for Por-COOH, Por-CH$_3$ and Por-H models after *OH formation. **f** The correlation between $\Delta G_{*OH}$, $d$-band center and ORR catalytic activity for Por-CH$_3$, Por-H, Por-COCH$_3$, Por-COOCH$_3$, Por-COOH models.

intrinsic catalytic performance. From the diagrams, the PDS of all porphyrin models is the final *OH release step, except for the Por-CN model (PDS of *OOH→*O). It is demonstrated that the OH$^-$ release from Co-N$_4$ sites demands a higher energy barrier for Por-CH$_3$, Por-H, Por-COCH$_3$ and Por-COOCH$_3$ models compared to the Por-COOH model, which in turn blocks further oxygen desorption. On the Por-COOH model, a reduced reaction energy barrier of 0.36 eV is discovered, meanwhile the Por-CN model requires a higher energy barrier of 0.39 eV to accelerate *OOH protonation conversion to *O as well. These results again reinforce that the carboxyl-substituted modification availably lowers the PDS energy barrier and elevates the oxygen electroreduction efficiency.

To better understand the activation and conversion mechanism of oxygen adsorbates, the electronic structure of orbital hybridization between Co 3$d$ orbital and O 2$p$ orbital was studied (Fig. 1d, e and Supplementary Fig. 10). On the basis of orbital spatial configurations, the σ-bond with strong energy splitting can be formed through orbital overlap between Co $d_{z^2}$ and *OH $p_z$ orbitals, while the twofold Co $d_{xz/yz}$ and *OH $p_{x/y}$ orbitals participate in hybridization to form relatively weak π-bonds[41,42]. Therefore, the $d$-$p$ hybridizations, including activated $d_{xz/yz}$ and $d_{z^2}$ orbitals co-determine the binding strength for

oxygen adsorbates. Therefore, it is expected that the local electronic structure on catalytic Co center was efficaciously managed to further weaken the *OH binding and realize the thermodynamic advantage of the desorption process.

As shown in Fig. 1e, under the substituent-induced electronic configuration tuning effect, the electron-donating methyl substituent is predicted to accept more electrons in the σ-bonding state, which is adverse to *OH desorption from Co-N$_4$ centers for the Por-CH$_3$ model. On the contrary, for the Por-COOH model, the distinct $p$ states of the oxygen specie, along with relatively poor hybridization between *OH $p_x/p_y$ and Co $d_{xz}/d_{yz}$ orbitals represent that the catalytic Co center furnishes a weaker interaction with *OH adsorbates than that of the original Por-H model. This further expedites the OH$^-$ conversion kinetics and balances the surface coverage by blocking adsorbed oxygen species, leading to the maximum possible electrocatalytic efficiency. The differential charge density maps upon *OH desorption on six porphyrin models are displayed to visualize the notable charge redistribution occurs between *OH species and Co centers (Supplementary Fig. 11). Also, the less electron transfer calculated for Por-COOH model stands for the weaker binding interactions of *OH intermediate on central Co atom[43]. Fig. 1f emphasizes that intrinsic

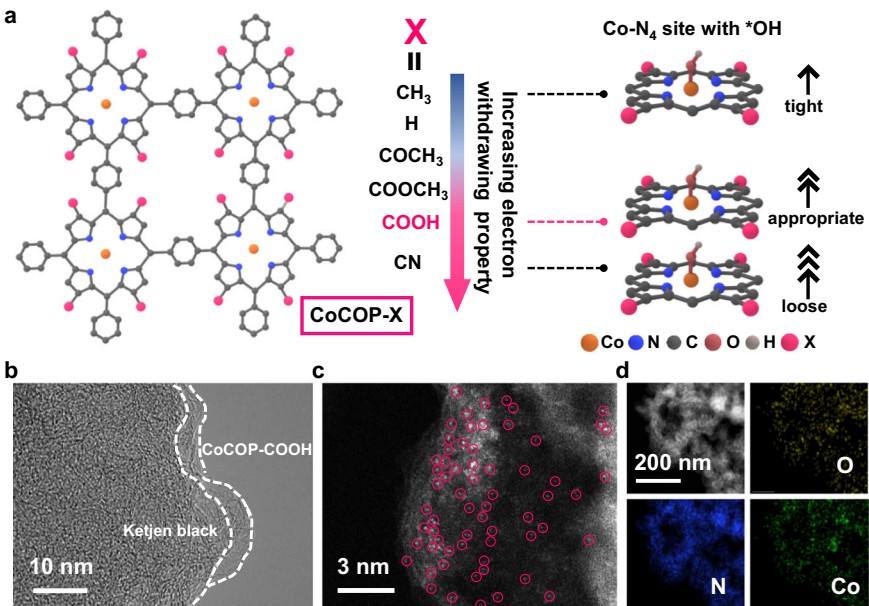

**Fig. 2 | Design principle and structural characterization of CoCOP-X@KB.**
**a** Schematic diagram of CoCOP-X analogs and adsorption strength illustration of adsorbed *OH intermediate with different Co-N$_4$ catalytic sites. **b** High-resolution TEM image of CoCOP-COOH@KB, the thin layer of CoCOP-COOH polymer with relatively low electronic conductivity was marked by white line. **c** Aberration-corrected HADDF-STEM image (isolated Co atoms are highlighted by pink circles) and **d** HAADF-STEM image with corresponding EDS elemental mapping of CoCOP-COOH@KB.

electrochemical activity of Por-X models depends on *OH adsorption energy and the position of *d*-band states relative to Fermi level. Specifically, the Por-CN model is not included since its excessive electron-withdrawing property changes the PDS, resulting in too weak or even unstable oxygen intermediate binding. Under the premise of maintaining the *OH desorption step as PDS, the correlation between ORR η and *d*-band center appears a Sabatier-type volcano plot along with the enhancement of the electron-withdrawing property on substituent. Notably, the electron-donating methyl substituent populates more σ-electrons around the Fermi level, leading to the *OH bonding reinforcement. With the increment of electron-withdrawing capability, the proportion of electrons in bonding orbital reduces, and the valence band gradually moves towards the Fermi level, thus the optimal local electronic microenvironment is finally achieved on the Por-COOH model. Thereby, the intrinsic reactivity emerges a linear dependence on the resultant *OH adsorption energy, and Por-COOH model with the weakest *OH binding strength greatly ameliorates *OH desorption rate, along with ORR thermodynamics and kinetics. These results express that the secondary sphere customization on Co porphyrin moieties is viable to accurately induce electron redistribution on the Co center, thus optimizing the bonding interaction with oxygen intermediates for effectively boosting oxygen reduction.

### Electrocatalyst synthesis and characterization

Motivated by the theoretical prediction, Co porphyrin-based polymer nanocomposite analogs (CoCOP-X@KB, X = CH$_3$, H, COCH$_3$, COOCH$_3$, COOH, CN) were prepared by mixing six secondary coordination sphere-substituted Co porphyrin polymers (CoCOP-X) with Ketjen black (KB), respectively (Fig. 2a). Taking advantage of the electron-withdrawing/donating properties alternation on the secondary coordination sphere substituent of the porphyrin moiety, the electronic configuration of central Co-N$_4$ and corresponding absorbate binding strength can be artfully manipulated.

The morphology and nanostructure of the as-synthesized CoCOP-X@KB nanocomposite analogs were carefully verified by scanning electron microscopy (SEM) and transmission electron microscopy (TEM). The pristine CoCOP-COOH polymer without KB discloses dense

and blocky micron-sized particle structures (Supplementary Fig. 12). It is discovered that, unlike CoCOP-COOH with smooth morphology, the surface of CoCOP-COOH@KB nanocomposite becomes rough after uniform loading on KB (Supplementary Figs. 13, 14), indicating the successful recombination of CoCOP-COOH polymer and the KB substrate. In addition, other CoCOP-X@KB nanocomposites also emerge alike morphologies and their energy-dispersive spectroscopy (EDS) mapping images demonstrate homogeneous distribution of corresponding elements as the secondary coordination sphere substituent changed (Supplementary Figs. 15–19). The TEM images illustrate that CoCOP-X@KB nanocomposites consist of interconnected nanoparticles, and no metallic Co particles are discovered (Supplementary Figs. 20–22). High-resolution TEM image (Fig. 2b) presents more details, i.e., CoCOP-COOH@KB reveals layers with several nanometer thickness that are evenly distributed on the KB substrate. Moreover, aberration-corrected high-angle annular dark-field, scanning transmission electron microscopy (HAADF-STEM) image of CoCOP-COOH@KB (Fig. 2c) shows a high density of bright dots which are related to the heavier Co elements with atomic dispersion, revealing that the Co metallic species are spatially isolated in Co-N$_4$ centers of CoCOP-COOH@KB. EDS mappings in HAADF-STEM (Fig. 2d) manifest homogeneous distribution of O, N and Co elements throughout the architecture of CoCOP-COOH@KB. The Co contents, measured by inductively coupled plasma optical emission spectroscopy (ICP-OES), were similar in all the catalysts, which guaranteed that the effect of molecular level microenvironment customization on ORR reactivity of CoCOP-X@KB catalysts can be explored independently (Supplementary Table 1).

To elucidate the formation of CoCOP-X@KB nanocomposite analogs, Fourier-transform infrared (FT-IR) spectroscopy was primarily carried out to establish the chemical structure (Supplementary Fig. 23). In the FT-IR spectrum of CoCOP-COOH@KB, the carbonyl group stretching band at about 1691 cm$^{-1}$ is slightly reduced relative to that of the 1,4-benzenedicarboxaldehyde monomer, and the remaining C=O band is attributed to the carboxyl substituent. After polymerization and recombination, the C=N stretching vibration band at around 1600 cm$^{-1}$ is still preserved in CoCOP-COOH@KB, which is similar to

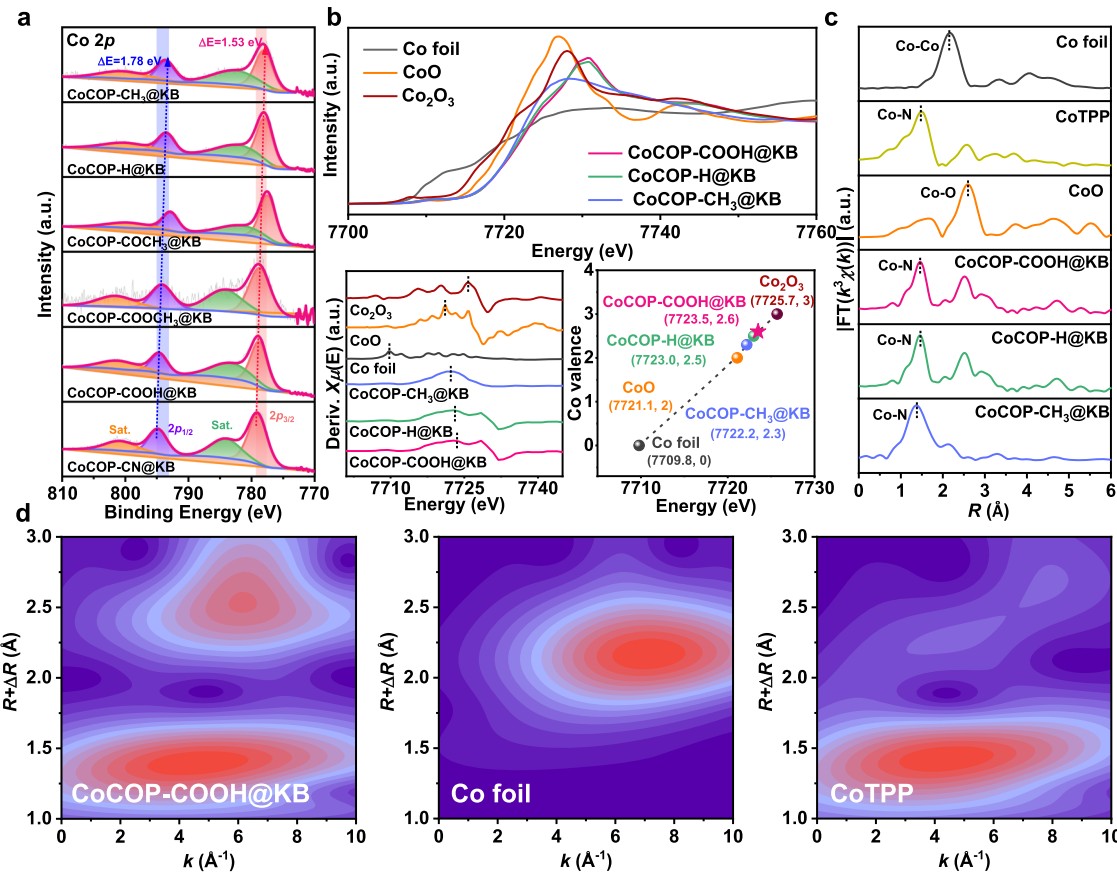

**Fig. 3 | Electronic states and atomic structure analysis of Co atoms in CoCOP-X@KB analogs. a** The high-resolution Co 2*p* XPS spectra of the as-synthesized CoCOP-X@KB analogs. **b** Normalized Co K-edge XANES spectra (up), first derivative of Co K-edge XANES spectra (bottom left), and average oxidation state determination by linear fitting calibration of CoCOP-COOH@KB, CoCOP-H@KB and CoCOP-CH₃@KB in comparison to Co foil, CoO, and Co₂O₃ control samples. **c** The Fourier transformed magnitude of the k³-weighted Co K-edge EXAFS spectra of CoCOP-COOH@KB, CoCOP-H@KB, CoCOP-CH₃@KB, Co foil, CoO, and CoTPP. **d** WT-EXAFS plots of CoCOP-COOH@KB, Co foil and CoTPP.

the carboxyl-substituted-pyrrole monomer, proving the conversion of the monomer to a polymer. The thermal stability of CoCOP-COOH on KB was confirmed by thermogravimetric analysis (TGA) and differential thermogravimetric analysis (DTG) (Supplementary Fig. 24). The representative TGA and DTG profiles of CoCOP-COOH@KB display peaks at temperatures of 200, 450 and 700 °C, which can be respectively ascribed to the substituent group decomposition, polymer skeleton decomposition and KB carbonization. Additionally, the weight loss under 100 °C may be due to the evaporation of solvents adsorbed in nanopores of CoCOP-COOH@KB. This further proves the successful incorporation of CoCOP-COOH polymer and KB. The X-ray diffraction (XRD) pattern of CoCOP-COOH@KB is predominated by the broad characteristic carbon (002) reflection of the KB substrate (Supplementary Fig. 25). It indicates the amorphous nature of the CoCOP-COOH polymer and well-dispersed Co atoms in CoCOP-COOH@KB, which is in accordance with HAADF-STEM results. The porous characteristics of CoCOP-X@KB analogs were assessed by nitrogen adsorption-desorption isotherms measurements (Supplementary Fig. 26 and Supplementary Table 2). The distinctly reduced nitrogen uptake (from 1340.0 m² g⁻¹ of KB to 538.5 m² g⁻¹ of CoCOP-COOH@KB) verifies the combination of CoCOP-COOH@KB, as the pores of the KB substrate are partially blocked by CoCOP-X polymers. The pore size distribution profiles (Supplementary Fig. 27) indicate that the pore structure is almost unchanged after the combination between CoCOP-X polymers and KB substrate, furthermore the skeletons of CoCOP-X@KB analogs are also well-maintained along with the substituent alternation.

Elemental chemical states and compositions of CoCOP-X@KB analogs were investigated by X-ray photoelectron spectroscopy (XPS). The high-resolution Co 2*p* spectra (Fig. 3a) for CoCOP-X@KB analogs can be deconvoluted into two main moieties at ~780.0 eV and 794.5 eV, which are assigned to Co 2*p*₃/₂ and Co 2*p*₁/₂ peaks, and the peaks at ~783.3 eV and 801.2 eV are ascribed to satellite peaks. Further, a maximum positive shift of about 1.78 eV can be found in CoCOP-X@KB analogs along with the enhanced electronegativity of the substituent, confirming the feasibility of secondary sphere customization strategy and the positively charged state of Co atom in CoCOP-COOH@KB (slight electron transfer from Co to carboxyl groups). Meantime, the XPS measurements identify the existence of corresponding elements in CoCOP-X@KB analogs, wherein substituents have been successfully incorporated in the polymer skeleton (Supplementary Figs. 28–33). The high-resolution N 1*s* XPS spectra affirm the dominance of pyrrolic nitrogen with a single sharp peak at a binding energy of about 397.5 eV, among CoCOP-X@KB analogs, besides CoCOP-CN@KB. The pronounced cyano nitrogen peak of CoCOP-CN@KB at about 396.8 eV verifies the presence of cyano groups in this compound material.

To discern the oxidation state and the coordination environment of the Co centers, X-ray absorption near-edge structure (XANES) and the Fourier-transform of the extended X-ray absorption fine structure (EXAFS) measurements and analyses were carried out. Initially, the XANES spectrum at Co K-edge is sufficiently sensitive for Co element valence state[44]. The Co K-edge XANES spectra (Fig. 3b) show that the absorption edge and white line intensity of CoCOP-COOH@KB,

CoCOP-H@KB and CoCOP-CH$_3$@KB are different from those of Co metal foil, CoO and Co$_2$O$_3$ control samples. The pre-edge at 7710 eV for CoCOP-COOH@KB, CoCOP-H@KB and CoCOP-CH$_3$@KB are triggered by the dipole forbidden 1s-to-3d transition, which normally represents the amount of unoccupied 3d states[45,46]. Thus, the higher pre-edge peak of CoCOP-COOH@KB stands for the increased extent of empty states in hybridized orbitals. Compared with CoCOP-CH$_3$@KB and CoCOP-H@KB counterparts, the Co K-edge XANES spectra edge energy (~7730 eV) for CoCOP-COOH@KB shows higher peak intensity, which represents a higher oxidation state of Co center in CoCOP-COOH@KB, consistent with the XPS results. To display the accurate oxidation states of Co atoms, the first derivative curve and average valence states determination by linear fitting calibration of the Co K-edge XANES were carried out. The first derivative curve exhibits that the $E_0$ value (calculated as the highest inflection point on the absorption edge) of CoCOP-COOH@KB (7723.5 eV) is situated between CoO (7721.1 eV) and Co$_2$O$_3$ (7725.7 eV). The linear fitting curves were obtained for CoCOP-COOH@KB, CoCOP-H@KB, CoCOP-CH$_3$@KB catalysts, and Co foil, CoO, Co$_2$O$_3$ serving as control samples. The linear fitting result suggests the different Co oxidation states of CoCOP-COOH@KB, CoCOP-H@KB and CoCOP-CH$_3$@KB ranging from +2 (CoO) to +3 (Co$_2$O$_3$). The average Co oxidation state of CoCOP-COOH@KB (+2.6) is the highest among three catalysts, which originates from the electron transfer from the Co-N$_4$ site to the carboxyl group. The EXAFS spectra further evidence that CoCOP-COOH@KB, CoCOP-H@KB and CoCOP-CH$_3$@KB nanocomposites only include isolated single Co atoms (Fig. 3c). The Fourier transforms of EXAFS for CoCOP-COOH@KB, CoCOP-H@KB and CoCOP-CH$_3$@KB are similar with that of Co tetraphenylporphyrin (CoTPP) with the dominant peak appearing at ~1.5 Å, which is attributed to the first shell coordination of Co-N scattering path. Importantly, the finger-printing signal peak of typical Co-Co interactions in the Co metal foil (~2.2 Å) is absent in the curve of CoCOP-COOH@KB, CoCOP-H@KB and CoCOP-CH$_3$@KB. According to the high resolutions in both R and k spaces, the wavelet-transforming (WT)-EXAFS analysis was employed to discriminate the backscattering atoms (Fig. 3d and Supplementary Fig. 34). Co foil furnishes an intensity maximum at (2.2 Å, 7.1 Å$^{-1}$), which is resulted from Co-Co coordination, yet the inexistence of intensity maximum at high k value in CoCOP-COOH@KB, CoCOP-H@KB and CoCOP-CH$_3$@KB clearly indicates that their Co centers do not directly bind to other Co atoms. Meanwhile, the intensity maximum at low k values (1.4 Å, 4.2 Å$^{-1}$) affirms their Co-N coordination, which is similar to CoTPP. In conclusion, the XANES and EXAFS spectra together with the HAADF-STEM results provide proof for the contention of atomic Co dispersion in nitrogen coordination environment of CoCOP-X@KB analogs.

## Electrochemical oxygen reduction performance

To evaluate the catalytic efficiency enhancement by virtue of secondary sphere microenvironment customization, the ORR electrochemical measurements for all CoCOP-X@KB catalysts were performed in a conventional three-electrode cell system. Noteworthy, apart from the electron-donating substituent-modified CoCOP-CH$_3$@KB catalyst, all electron-withdrawing substituent-modified CoCOP-X@KB catalysts possess improved ORR performance compared to the pristine CoCOP-H@KB counterpart. To estimate the basic activity indicators of each electrocatalyst, the onset potential ($E_{onset}$) and half-wave potential ($E_{1/2}$) were gathered by linear sweep voltammetry (LSV) polarization curves (Fig. 4a). Interestingly, as far as $E_{onset}$ and $E_{1/2}$ are concerned, an evident tendency is that the absolute values of CoCOP-X@KB catalysts exhibit a volcano correlation along with the subtle electronegativity enhancement of the substituent (Fig. 4b). Specially, CoCOP-COOH@KB owns optimal catalytic performance with the highest $E_{onset}$ and $E_{1/2}$ of 0.95 V and 0.86 V (vs. RHE), which are much more positive than those of CoCOP-H@KB (0.85 V, 0.75 V) and CoCOP-CH$_3$@KB (0.85 V, 0.73 V) counterparts (Fig. 4c). The related

ORR parameters, kinetic current density ($J_k$) and mass activity ($J_m$) for electron-withdrawing substituent-modified CoCOP-COOH@KB, electron-donating substituent-modified CoCOP-CH$_3$@KB and pristine CoCOP-H@KB counterparts are collected (Fig. 4d and Supplementary Fig. 35). The $J_k$ for CoCOP-COOH@KB electrode reaches 12.5 mA cm$^{-2}$ at 0.8 V, which is 62.5- and 25.0-fold promoted than those of CoCOP-CH$_3$@KB (0.2 mA cm$^{-2}$) and CoCOP-H@KB electrodes (0.5 mA cm$^{-2}$), respectively. Meantime, the $J_m$ for CoCOP-COOH@KB (54.9 A g$^{-1}$) also achieves an order of magnitude increase over CoCOP-CH$_3$@KB (2.4 A g$^{-1}$) and CoCOP-H@KB (3.7 A g$^{-1}$), demonstrating that carboxyl group ensures a well-managed electron-efficient Co center, along with favorable ORR dynamics and kinetics for CoCOP-COOH@KB.

More ORR properties of CoCOP-X@KB catalysts were derived from the catalytic Tafel plots after mass-transport limitation correction (Fig. 4e). In the low $\eta$ region, CoCOP-COOH@KB reveals the smallest Tafel slope (41.1 mV dec$^{-1}$) among CoCOP-X@KB, implicating the highest transfer coefficient of CoCOP-COOH@KB for ORR electrocatalytic kinetics. These results support the theoretical predictions that the carboxyl-substituted modification effectively customizes the desorption behaviors of *OH reaction intermediates thus accelerating the sluggish oxygen electrocatalysis. Apart from the ORR catalytic activity, the rotating ring-disk electrode (RRDE) measurement was utilized to evaluate the HO$_2^-$ yield and the number of electrons transferred as a function of applied potential for all six catalysts (Supplementary Figs. 36–41). Koutecký–Levich (K-L) plots, i.e., $J^{-1}$ vs. $\omega^{1/2}$, were also obtained from LSV curves to determine the catalytic selectivity. The electron transfer number per O$_2$ calculated from the K-L equation increased from 3.37 for CoCOP-H@KB to 3.49 for CoCOP-COOH@KB, which coincides with the results from RRDE voltammetry, confirming the significant selectivity promotion of CoCOP-COOH@KB towards the four-electron ORR pathway.

The electrochemical active surface area (ECSA) was compared by estimating the electrochemical double-layer capacitance ($C_{dl}$), which was derived from the cyclic voltammetry (CV) results in the potential range of 1.0 V to 1.1 V (Supplementary Fig. 42). The $C_{dl}$ value of CoCOP-COOH@KB (12.55 mF cm$^{-2}$) is slightly larger than those of the CoCOP-CH$_3$@KB (9.25 mF cm$^{-2}$) and CoCOP-H@KB (9.28 mF cm$^{-2}$) counterparts. The resultant enlarged ECSA of CoCOP-COOH@KB elaborates the greater accessible active surface and abundant micro-mesopore structures, which are more favorable for effective charge and mass transfer in catalysis.

Besides, long-term stability is also a pivotal indicator to appraise the ORR performance for practical applications of electrocatalysts. CoCOP-COOH@KB shows good long-term stability with a minimal $E_{1/2}$ decline after 5000 continuous CV cycles (Supplementary Fig. 43). Chronoamperometric (current-time) responses were obtained on CoCOP-COOH@KB and commercial Pt/C at 0.4 V (Supplementary Fig. 44). During the steady-state potentiostatic measurements, the current density of CoCOP-COOH@KB retains up to 97.1% after about 10 h of continuous operation, while the initial current density of Pt/C decays to 83.3% under the same condition, representing the eminent durability for the CoCOP-COOH@KB catalyst. CoCOP-COOH@KB also shows better methanol tolerance than Pt/C (Supplementary Fig. 45), further manifesting its great chemical stability.

Considering the satisfactory intrinsic catalytic activity, the CoCOP-COOH@KB catalyst was loaded on carbon paper to prepare the air cathode for testing in ZAB configuration, with the Zn foil anode and 6 M KOH aqueous electrolyte solution, while the Pt/C-based primary ZAB was used for comparison (Supplementary Fig. 46a, b). According to polarization plots and derived power density profiles (Fig. 4f), CoCOP-COOH@KB reaches a peak power density of 239.6 mW cm$^{-2}$ at a current density of 320.5 mA cm$^{-2}$, surpassing the Pt/C-based ZAB (166.9 mW cm$^{-2}$ at 243.5 mA cm$^{-2}$). Particularly, the discharge performance of CoCOP-COOH@KB is better by about 20% than that of Pt/C at high current density (>110 mA cm$^{-2}$) due to its

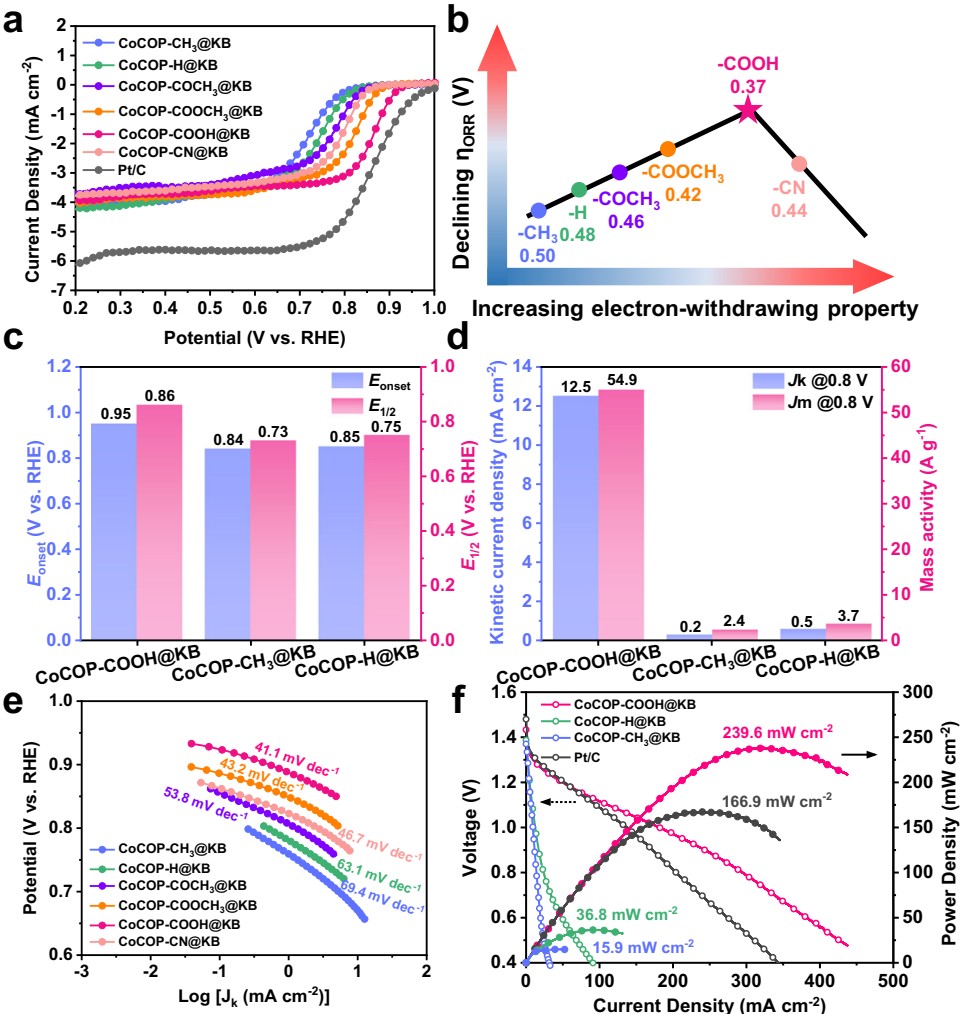

**Fig. 4 | Experimentally validated Sabatier volcano plot of CoCOP-X@KB catalysts and performance analysis. a** LSV curves of CoCOP-X@KB catalysts in O$_2$-saturated 0.1 M KOH solution at a rotation speed of 1600 rpm. **b** Relationship between the electron-withdrawing property of substituents on the porphyrin secondary coordination sphere and corresponding ORR activity of CoCOP-X@KB catalysts. The contrast between CoCOP-COOH@KB, CoCOP-CH$_3$@KB and CoCOP-H@KB for **c** $E_{onset}$ and $E_{1/2}$ at 0.8 V, **d** $J_k$ and $J_m$ at 0.8 V. **e** Tafel curves of CoCOP-X@KB catalysts. **f** Discharging polarization curves and the corresponding power density plots of CoCOP-COOH@KB, CoCOP-H@KB, CoCOP-CH$_3$@KB-based and Pt/C-based ZABs.

fast charge and mass transfer kinetics. The specific capacity of CoCOP-COOH@KB-based ZAB is 807.1 mAh g$_{Zn}^{-1}$ (corresponding to the specific energy of ~969 Wh kg$_{Zn}^{-1}$) at 50 mA cm$^{-2}$ (Supplementary Fig. 46c), achieving almost 99% utilization of Zn, exceeding that of commercial Pt/C (779.2 mAh g$^{-1}$ with 95% utilization, ~943 Wh kg$_{Zn}^{-1}$). Additionally, the galvanostatic discharge processes (Supplementary Fig. 46d) at various current densities (5, 10, 20, 50, and 100 mA cm$^{-2}$) show that the assembled CoCOP-COOH@KB-based ZAB delivers higher discharge voltage than the commercial Pt/C equipped one. The output voltage of the ZAB with CoCOP-COOH@KB almost recovers when the current density returns to 5 mA cm$^{-2}$, implying its improved rate performance and recoverability, which can be ascribed to the accelerated ORR kinetics, fast charge/ion diffusion and boosted mass transfer of CoCOP-COOH@KB electrode. The rechargeability and cyclic durability of CoCOP-COOH@KB and Pt/C counterpart were then tested (Supplementary Fig. 46e). Note that the CoCOP-COOH@KB catalyst demonstrates improved cycling stability and can be stably charged and discharged over 300 hours at a current density of 25 mA cm$^{-2}$. After long-term operation, the negligible voltage-gap increase of CoCOP-COOH@KB-based ZAB indicates its excellent durability, which outperforms that of the commercial Pt/C catalysts.

## Verification of the ORR mechanism

In order to study the underlying correlation between electronic microenvironment around the Co center and corresponding intrinsic reactivity, scanning electrochemical microscopy (SECM) was firstly employed to in situ visualize the local ORR activity of CoCOP-COOH@KB, CoCOP-H@KB and CoCOP-CH$_3$@KB. Peculiarly, the SECM technology is more inclined to probe the real-time electrocatalytic process and quantify the number of active sites involved in an electrochemical reaction due to its in situ micro-/nanoscale observation of instantaneous product generation at the catalysts interfaces[47,48]. The schematic of measuring protocol for ORR is illustrated in Fig. 5a, the SECM technology comprises a Pt ultra-micro probe electrode (UME) and an Au substrate microdisk concave electrode loaded with the catalyst powder (Supplementary Fig. 47). The local electrochemical behaviors of CoCOP-COOH@KB, CoCOP-H@KB and CoCOP-CH$_3$@KB catalysts are mapped in tip-generation/substrate-collection (TG/SC) mode, where a certain high potential is applied to the probe electrode to form oxygen molecule for substrate to initiate ORR process. As delineated in Fig. 5b–d, the obtained SECM scanning images of three catalysts clearly outline the conductive regions and exhibit apparent feedback currents from both probe ($I_p$) and substrate ($I_s$). The SECM scanning images of $I_p$ and $I_s$

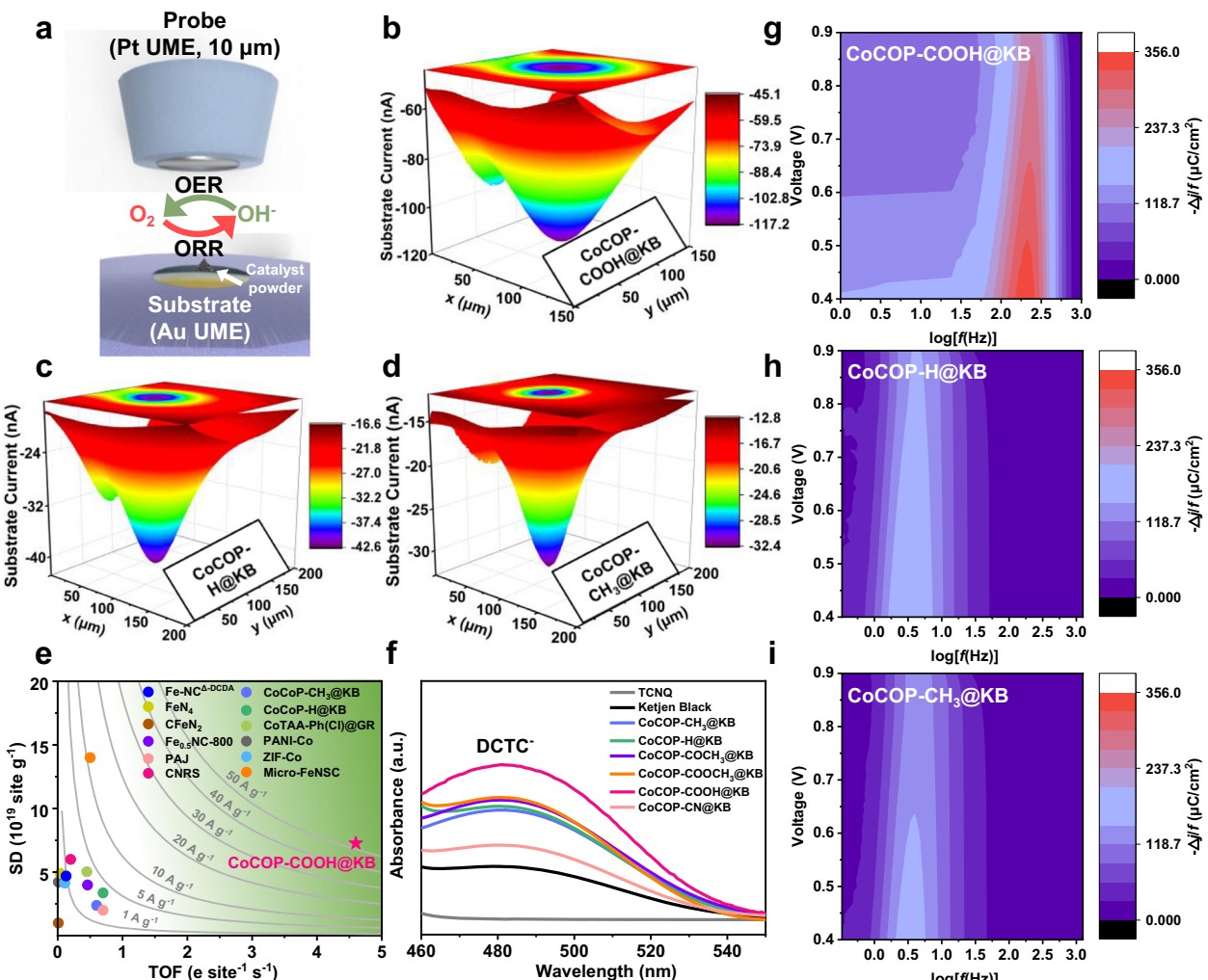

**Fig. 5 | Origin of the activity volcano over CoCOP-X@KB catalysts induced by secondary sphere microenvironment customization. a** Setup used for a typical SECM investigation in the TG/SC mode. OER, oxygen evolution reaction. TG/SC mode SECM images at $E_s$ = 0.0 V of **b** CoCOP-COOH@KB, **c** CoCOP-H@KB and **d** CoCOP-CH₃@KB. **e** The ORR isoactivity map derived from the value of surface SD and TOF at 0.8 V for CoCOP-COOH@KB, CoCOP-CH₃@KB, CoCOP-H@KB and recently reported advanced single-atom catalysts in literature. **f** Representative UV-Vis spectra of TCNQ solutions in acetonitrile with KB and CoCOP-X@KB analogs. VF-SWV colormaps of **g** CoCOP-COOH@KB, **h** CoCOP-H@KB and **i** CoCOP-CH₃@KB electrodes.

corresponding to various applied substrate potentials ($E_s$, from 0.0 V to 0.9 V) of three catalysts were collected, respectively (Supplementary Fig. 48–50). It can be seen that the $I_p$ and $I_s$ are observably enhanced along with the increase of given bias on substrate, which is consistent with the LSV results. Both the detected $I_p$ and $I_s$ of CoCOP-COOH@KB are explicitly greater than those of CoCOP-H@KB and CoCOP-CH₃@KB over the whole ORR potential range. Further, the value of maximum $I_s$ is considered to visually reflect the local catalytic reactivity (Supplementary Fig. 51), notably, for CoCOP-COOH@KB (117.2 nA) at $E_s$ = 0.0 V, over 2.7 and 3.7-fold enhancement of $I_s$ can be recorded compared to that of CoCOP-H@KB (42.6 nA) and CoCOP-CH₃@KB (32.4 nA). This implies that carboxyl-substituted modification renders dramatic improvement on electronic conductivity and electrocatalytic activity relative to other counterparts, as confirmed by more intensive SECM feedback with OH⁻/O₂ as primary redox mediator.

To in situ quantify the active sites factually participating in the oxygen electroreduction reaction and comprehend the catalytic kinetics on active sites, the surface-interrogation SECM (SI-SECM) technique was employed[49–51]. Briefly, the molecular probe FcMeOH⁺ electrogenerated from FcMeOH at the UME diffuses to the substrate electrode, which is subsequently titrated by reactive Co(II) sites on

the substrate electrode under given potentials, thereby recording the positive current response caused by the FcMeOH⁺/FcMeOH transition (Supplementary Fig. 52). The resultant FcMeOH oxidation feedback currents at the tip probe UME finally attains a steady-state as the depletion of all Co(II) active species, and the integrated tip-collected electric charge quantity ($Q_{tip}$) can be calculated to determine the number of participant active sites (Supplementary Fig. 53). Furthermore, the SD is acquired through normalizing the active sites number with ECSA indirectly obtained by the $C_{dl}$ method (Supplementary Fig. 54). Note that the TOF can be also transformed according to the SI-SECM-determined SD to provide a more dependable and less restrictive assessment on intrinsic catalyst performance. The ORR isoactivity maps were derived from the values of SD and TOF at 0.8 V to compare CoCOP-COOH@KB, CoCOP-H@KB and CoCOP-CH₃@KB catalysts with recently reported Pt-free catalysts (Fig. 5e and Supplementary Table 3). As can be seen, CoCOP-COOH@KB possesses higher SD of $7.3 \times 10^{19}$ sites g⁻¹, with an improvement of more than twofold when contrasted to CoCOP-H@KB ($3.4 \times 10^{19}$ sites g⁻¹) and CoCOP-CH₃@KB ($2.4 \times 10^{19}$ sites g⁻¹). Additionally, CoCOP-COOH@KB not only supplies denser catalytic sites but also offers an improved TOF value of individual atomic Co. The TOF value of CoCOP-COOH@KB is as high as 4.6 e s⁻¹ site⁻¹, which is more than sevenfold that of CoCOP-H@KB

(0.7 e s$^{-1}$ site$^{-1}$) and CoCOP-CH$_3$@KB (0.6 e s$^{-1}$ site$^{-1}$). The ORR SD-TOF isoactivity maps further emerge that CoCOP-COOH@KB combines a relatively high SD and excellent TOF, thus resulting in remarkable SI-SECM-determined $J_m$ over 50 A g$^{-1}$. In comparison with recently reported pyrolysis-free atomically dispersed M-N-C catalysts[49], or even carbonaceous catalysts[52–56], CoCOP-COOH@KB achieves better site-specific activity for ORR, which is found to approach the top of SD-TOF isoactivity map. This result emphasizes the advantages of the secondary sphere customization strategy featuring with favorable catalytic microenvironment regulation on Co-N$_4$ center, along with superb ORR thermodynamic and kinetic control.

To quantify the electron outbound propagation ability of catalysts and investigate the interfacial charge transfer mechanism, we utilized 7,7,8,8-tetracyanoquinodimethane (TCNQ) as electron-accepting molecule and captured the α,α-dicyano-p-toluoylcyanide anion (DCTC$^-$) formed during the reaction (Supplementary Fig. 55a). In brief, based on the oxygen decay chemistry of TCNQ$^{2-}$, TCNQ$^{2-}$ react with O$_2$ to form a dioxetane intermediate, which in turn loses a cyanate ion to obtain DCTC$^-$. The UV-Vis spectra of DCTC$^-$, which is chemically generated by TCNQ solutions catalyzed with KB and CoCOP-X@KB analogs are shown in Fig. 5f. As opposed to other weak absorptions around 480 nm, the TCNQ solution with CoCOP-COOH@KB reveals an intense absorption. Meanwhile, the original TCNQ solution appears greenish due to the adsorption of infrared/near-infrared light by TCNQ radical anions in acetonitrile solution[57]. After the thorough exposure to different catalysts, unlike the dark green TCNQ$^{2-}$ anion solution, the reaction between TCNQ and CoCOP-COOH@KB produces absolutely different tawny DCTC$^-$ solutions due to the subsequent TCNQ$^{2-}$ oxidation (Supplementary Fig. 55b). Hence, the enhanced catalytic performance of CoCOP-COOH@KB is closely related to its fast electron transfer kinetics during ORR process.

Given this, we carried out ultraviolet photoemission spectroscopy (UPS) measurement to survey the electronic structure and work function of CoCOP-COOH@KB, CoCOP-H@KB and CoCOP-CH$_3$@KB. CoCOP-COOH@KB shows a slight shift in the secondary electron cutoff region to higher binding energy (Supplementary Fig. 56). The lower work function of CoCOP-COOH@KB (4.38 eV) compared to those of CoCOP-H@KB (4.48 eV) and CoCOP-CH$_3$@KB (4.56 eV) manifests its lower energy barrier required to activate an outward-directed electron transfer to adsorbed oxygen species, which is more conducive to subsequent ORR steps. The electrochemical impedance spectroscopy (EIS) measurements were also displayed to explore the charge transfer resistance and electrode kinetics (Supplementary Fig. 57 and Supplementary Table 4). The equivalent circuit model was set to be the combination of solution resistance (R$_s$), charge transfer resistance (R$_{ct}$), constant phase element (CPE) and Warburg resistance for ionic diffusion processes (W). The smaller intercept with the real axis in the high frequency region, the relatively lower value of R$_{ct}$, along with higher slope at low frequency region for CoCOP-COOH@KB attests its reduced Ohmic resistances and enhanced ion diffusion in electrolyte, uncovering its improved interfacial electron transfer kinetics. Besides, the average fluorescence lifetime of CoCOP-COOH@KB (0.38 ns) is shorter than other five CoCOP-X@KB analogs, again indicating the improved electron transfer property (Supplementary Fig. 58).

In view of the importance of nonequilibrium electron transfer rates for the state-of-the-art heterogeneous catalysts, variable-frequency square wave voltammetry (VF-SWV) was proposed to interpret the reversible redox behavior on Co-N$_4$ sites. The generation and reversible conversion of high-valent metal intermediates are generally regarded as a key determining descriptor impacting the ultimate ORR performance. The interfacial and interlayer charge transfer were directly electrochemically visualized through kinetic colormaps obtained by the VF-SWV method[58–60]. The mappings of VF-SWV study for CoCOP-COOH@KB, CoCOP-H@KB, and CoCOP-CH$_3$@KB counterparts were attained by plotting the frequency-normalized current density -Δj/f responses against SWV log(f) (Fig. 5g–i). Indeed, the CoCOP-COOH@KB electrode shows a higher frequency response signal at the log(f) of 2.3 than CoCOP-H@KB and CoCOP-CH$_3$@KB (log(f) of 0.5), respectively. It should be noted that the intensity of the current response signal for CoCOP-COOH@KB is ca. 2 times greater compared to the other two counterparts, affirming the rapid electron outbound propagation and fast high-valent metal redox transformation.

For ascertaining the intermediates absorbed at Co sites during ORR and determining the dynamic evolution behavior on catalyst surface, progressive testing techniques incorporating electrochemical measurements and in situ characterization approaches were used to achieve a real-time analysis. The operando attenuated total reflectance surface-enhanced infrared absorption spectroscopy (ATR-SEIRAS) presents that the peak at 820 cm$^{-1}$ can be attributed to the O-O stretching in absorbed O$_2$ species (Fig. 6a)[61]. Further, the intensity of O-O stretching band rapidly decreases by varying the cathodic potential stepwise from 1.0 V to 0.6 V for CoCOP-COOH@KB (Fig. 6b), manifesting the consumption of dissolved oxygen.

In addition, in situ electrochemical Raman spectroscopy measurements under a series of applied bias voltages on CoCOP-COOH@KB, CoCOP-H@KB, CoCOP-CH$_3$@KB were carried out and the site transformation during ORR process was probed simultaneously (Fig. 6c and Supplementary Fig. 59). The prepared CoCOP-COOH@KB-based working electrode exhibits a peak at 1220 cm$^{-1}$ belonging to the oxygen molecule adsorption on the Co center, which decreases with the increase of $\eta$, representing the occurrence of ORR process and the descending concentration of dissolved oxygen. Shifting the potential to 0.9 V to ensure that ORR has occurred, two bands can be observed at 765 and 1550 cm$^{-1}$, which is assigned to the ν(Co-OH) and ν(Co-OOH) vibrations, respectively[62–64]. While further negatively alternating the potential, the band for Co-OOH gradually vanishes and the intensity of the Co-OH peak decreases due to the conversion of *OH to OH$^-$ species on the Co-N$_4$ center. In particular, the spectral pattern of CoCOP-COOH@KB is almost identical when the potential returns from 0.3 V to 1.1 V, demonstrating the great stability and reversibility of CoCOP-COOH@KB. For CoCOP-H@KB electrode (Supplementary Fig. 59a), the peak at around 600 and 690 cm$^{-1}$ were assigned to the symmetric stretching of ν(Co-OOH) and ν(Co-O), respectively. As the potential decreases, ν(Co-OOH) signal weakens when the ν(Co-O) increases as the potential reaches 0.5 V, which is originated from the blockage of Co centers by subsequent firmly adsorbed *OH species. Moreover, the peaks at 780 cm$^{-1}$ for CoCOP-CH$_3$@KB electrode (Supplementary Fig. 59b), which represents the ν(Co-OH) signal, further confirms the overmuch oxygen binding strength on electron-donating group modified Co-N$_4$ center. Based on the results presented, we conclude that the porphyrin-based Co-N$_4$ centers normally possess high affinity toward O$_2$ species, the *OH species acts as the key intermediate and the final *OH desorption step is the PDS for ORR on original porphyrinic Co-N$_4$ active sites. The carboxyl substituent accelerates the ORR kinetics on CoCOP-COOH@KB by relieving the overmuch binding strength, which is in good agreement with DFT calculations. These findings elucidate the elementary processes over CoCOP-COOH@KB catalysts, and the corresponding ORR mechanism is proposed in Supplementary Fig. 60.

The electron spin configuration and unpaired electrons on Co ions within CoCOP-COOH@KB, CoCOP-H@KB and CoCOP-CH$_3$@KB counterparts were probed by the zero-field cooling temperature-dependent magnetic susceptibility and ex situ cryo (100 K) electron paramagnetic resonance (EPR) spectroscopy. The average number of the unpaired Co 3$d$ electrons acquired from the magnetic susceptibility plots (Fig. 6d) decreases along with the electron-withdrawing property of secondary sphere substituent increases. This result suggests that the strong electron-withdrawing/donating effect induces

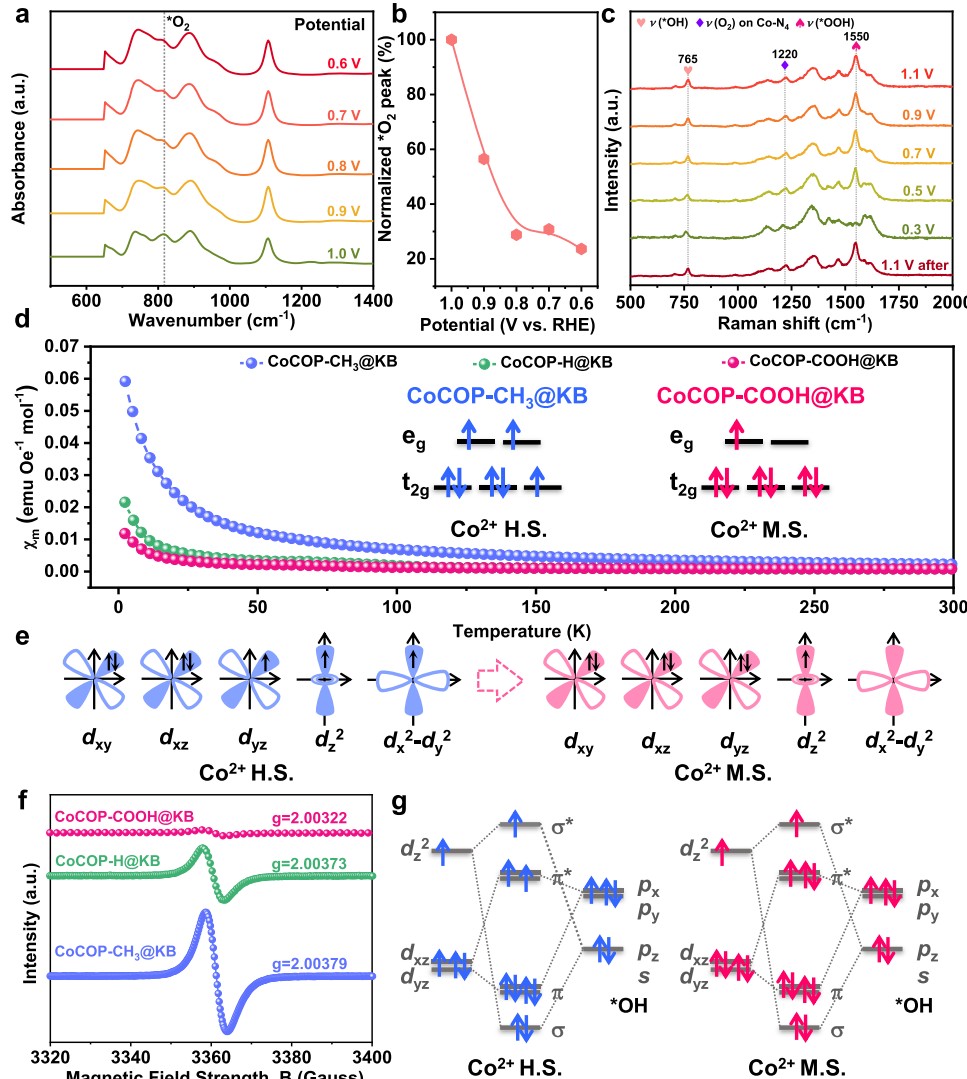

**Fig. 6 | The in situ electrochemical spectroscopic characterizations and schematic diagram of potential determining step transition. a** Operando ATR-SEIRAS spectra for ORR at 500–2000 cm$^{-1}$, **b** potential-dependent O-O stretching peak intensity, **c** potential-dependent in situ Raman spectra during ORR process over CoCOP-COOH@KB catalysts in 0.1 M KOH solution. **d** Magnetic susceptibility of CoCOP-COOH@KB, CoCOP-H@KB and CoCOP-CH$_3$@KB catalysts. M.S. represents medium spin, H.S. represents high spin. **e** 3$d$ electron occupations in the M.S. and H.S. states of Co$^{2+}$ in CoCOP-X@KB. **f** Low-temperature EPR spectra recorded at $T = 100$ K of CoCOP-COOH@KB, CoCOP-H@KB and CoCOP-CH$_3$@KB catalysts. **g** The orbital interactions between Co centers with different spin states and the potential determining *OH intermediates.

the Co 3$d$ electronic spin state transition, and the Co 3$d$ shells possesses different electronic configurations, which are high spin state with $d_{xy}2d_{xz}1d_{yz}1d_z^21d_{x^2-y^2}1$ and medium spin state with $d_{xy}2d_{xz}2d_{yz}2d_z^21$ (Fig. 6e). Further, the shape and intensity of EPR profile are impacted by the lone pair electrons onto the structural transition (Fig. 6f). The EPR spectrum shows an obvious signal at g ≈ 2.00 for CoCOP-CH$_3$@KB and CoCOP-H@KB, which is attributed to a typical high spin Co$^{2+}$ signal. As the enhancement of the electron-withdrawing property on substituent, the spin structure of catalytic Co sites changes from high spin (CoCOP-CH$_3$@KB and CoCOP-H@KB) to medium spin (CoCOP-COOH@KB). The EPR results indicates the weakened O$_2$ adsorption affinity of CoCOP-COOH@KB, which is in agreement with the theoretical calculation. Thermodynamically, the relatively weak adsorption of *OH permits the rapid final *OH release, and the $d_{x^2-y^2}$ and $d_{xy}$ orbitals are neglected due to their negligible interactions with oxygen intermediates. Compared with high spin Co centers within CoCOP-CH$_3$@KB, the decreased proportion of electrons in bonding orbital of medium spin Co$^{2+}$ within CoCOP-COOH@KB leads to an alleviative Co/*OH interaction (Fig. 6g), which is more favorable for propelling ORR cycles.

## Discussion

In summary, by analyzing structures-reactivity relation of secondary sphere-substituted porphyrin models, the oxygen intermediates adsorption energy descriptor can be well captured with the microenvironment of Co porphyrin-based polymer catalysts, which verifies the Sabatier volcano correlation and affords the prediction of ORR $\eta$. The descriptor works effectively to describe the mechanism-based performance optimization tendency that electron-withdrawing substituents reduce the proportion of electrons in bonding orbital to weaken the vital *OH intermediate binding strength on the active Co center, thereby facilitate ORR thermodynamics and kinetics. Based on this descriptor, we present a microenvironment customization strategy characterized by tuning electron-withdrawing/donating property on porphyrin secondary sphere substituent to customize O$_2$ adsorption ability and reactivity on the Co-N$_4$ site. The Co porphyrin-based

polymer analogs were prepared and the optimal carboxyl group-substituted CoCOP-COOH@KB catalyst resided near the volcano top with the favorable kinetic characteristics towards ORR in terms of high $E_{1/2}$ (0.86 V), great $J_k$ (12.5 mA cm$^{-2}$) and $J_m$ (54.9 A g$^{-1}$). The ZAB equipped with CoCOP-COOH@KB catalyst shows great peak power density (239.6 mW cm$^{-2}$), large specific capacity (807.1 mAh g$_{Zn}^{-1}$) and long-term cycle life (over 300 hours). Systematic approach based on in situ SECM technique verifies that CoCOP-COOH@KB owns improved intrinsic activity on individual atomic Co center with greatly enhanced TOF value (4.6 e s$^{-1}$ site$^{-1}$) and accessible SD (7.3 × 10$^{19}$ sites g$^{-1}$). Electrochemical VF-SWV method proves that CoCOP-COOH@KB kinetically favors the activation and outward transport of charges to the oxygen adsorbates. In situ electrochemical spectroscopic characterizations investigate the underlying evolving mechanism on Co sites and affirm that *OH desorption is the PDS of oxygen electroreduction process.

## Methods

### Computational details

Density functional theory (DFT) calculations were performed by employing the Vienna ab initio simulation package (VASP)[65] utilizing the projector augmented wave (PAW) potentials with a planewave cutoff energy of 450 eV[66,67]. The exchange-correlation functional[68] was described by generalized gradient approximation (GGA) functional of Perdew, Burke, and Ernzerhof (PBE) with grimme's semiempirical DFT-D3 dispersion correction to describe the van der Waals (vdW) interactions[69]. To prevent the interaction between two adjacent layers, a vacuum layer of 15 Å was applied for all of the surface models. We applied a gamma-centered k-point grid of 3 × 3 × 1 to sample the Brillouin zones for all structures. The convergence criteria of electronic energies were 10$^{-5}$ eV, and the atomic force was 0.03 eV/Å for all calculations. To study the ORR reaction pathways, the *OOH, *O, and *OH absorbed intermediates were optimized by DFT calculations to obtain the energy of each structure. The Gibbs free energy (ΔG) diagram of the ORR was calculated using the computational hydrogen electrode model proposed by Nørskov et al.[37], where the free energy of (H$^+$ + e$^-$) under standard conditions is equal to the value of 1/2 H$_2$. Free energy is calculated by the formula:

$$\Delta G = \Delta E + \Delta ZPE - \Delta TS + \Delta G_U + \Delta G_{PH} \quad (1)$$

where e is the total energy of the system, ZPE is the zero-point energy, and S is the entropy.

The Zero-point-energy (ZPE) and entropy corrections were performed through frequency calculations. *U* is the applied potential, and $\Delta G_{PH}$ is considered as the correction free energy of H$^+$. The overall reaction of ORR occurring in the alkaline conditions is: $O_2 + 2H_2O + 4e^- = 4OH^-$. The ORR proceeds through the following elementary steps, which are generally used to prove the ORR electrocatalytic process on active sites:

$$O_2\,(g) + H_2O\,(l) + e^- + ^* \rightarrow\, ^*OOH + OH^- \quad (2)$$

$$^*OOH + e^- \rightarrow\, ^*O + OH^- \quad (3)$$

$$^*O + H_2O\,(l) + e^- \rightarrow\, ^*OH + OH^- \quad (4)$$

$$^*OH + e^- \rightarrow\, OH^- + ^* \quad (5)$$

where * represents the active site on catalytic surface, (*l*) and (*g*) stand for liquid and gas phases, respectively.

### Materials and reagents

Pyrrole, 3-methylpyrrole, pyrrole-3-carboxylic acid, 1,4-phthalaldehyde (BDA), trifluoroacetic acid (TFA), nitrobenzene (NBZ) were purchased from Energy Chemical Co. Ltd. 3-Cyanopyrrole, 3-acetylpyrrole and methylpyrrole-3-carboxylate were bought from Leyan (Shanghai, China). Potassium hydroxide (KOH) was purchased from Sinopharm Chemical Reagent Co. Ltd (China). Propionic acid was bought from Alfa Aesar. KB was purchased from Cabot Corporation. Carbon paper was obtained from Hesen Electric Inc. Shanghai. Zinc foil was obtained from Tengfeng Metal Co., Ltd. All the reagents and solvents are analytical grade and used without further purification.

### Synthesis of CoCOP-COOH polymer

First, BDA (54.7 mg, 0.4 mmol) was dissolved into a mixture solution with 12 mL propionic acid, 80 μL TFA, and 250 μL NBZ under ultrasonic treatment for 30 min to form a homogenous suspension. After that, 5 mL propionic acid solution of pyrrole-3-carboxylic acid was added, then the as-obtained mixture was kept at 140 °C under constant magnetic stirring condition for 2 h to form a porphyrin-based polymer skeleton. The cobalt acetate equivalent to porphyrin moiety was dispersed in 8 mL propionic acid and added dropwise to the above solution under stirring. The mixture was kept at 140 °C for another 12 h, then cooled to room temperature (-25 °C) to provide a dark suspension. The suspension was filtrated, then washed with chloroform, deionized water and ethanol to remove unreacted reagents and porphyrin molecules. After vacuum drying at 60 °C overnight, CoCOP-COOH polymer was obtained.

### Synthesis of CoCOP-COOH@KB nanocomposite

Typically, 40 mg CoCOP-COOH polymer and 20 mg KB were well mixed by grinding in agate mortar containing agate balls (diameter of 0.6 and 0.8 mm) and 3 mL methanol under air atmosphere. The grinding bowl was later set into the ball mill and revolved for 6 h at 300 rpm. After separating the grinding beads, the obtained CoCOP-COOH@KB was washed with deionized water and ethanol for three times. After vacuum drying at 60 °C overnight, CoCOP-COOH@KB nanocomposite was obtained.

### Synthesis of other CoCOP-X@KB nanocomposites

The other CoCOP-X@KB nanocomposites were synthesized and purified using similar procedures as CoCOP-COOH@KB, but replacing original pyrrole by five different substituted-pyrrole to prepare corresponding secondary sphere-substituted-porphyrin polymer nanocomposites. The subsequent preparation processes were similar to those for CoCOP-COOH@KB.

### Instrumentations

SEM measurements were performed via a scanning electron microscope (JEOL JSM-7900F). TEM images, HRTEM images, SAED patterns, EDS and HAADF-STEM were obtained by employing a scanning transmission electron microscope (JEOL, JEM-2100F). The AC HAADF-STM measurement was performed by a scanning transmission electron microscope (Titan Cubed Themis G2 300). FT-IR spectra were recorded on a Thermo Scientific Nicolet 6700. The nitrogen adsorption-desorption measurements were performed on an Autosorb iQ (Anton-Paar China), and the specific surface areas were obtained by the BET model. XRD patterns were obtained by a Bruker 102 D8Discover 25 X-ray diffractometer. XPS spectra were recorded on a Kratos AXIS Ultra using monochromatized Al$_\alpha$-radiation. UPS spectra were recorded on a Thermo ESCALAB 250XI PHI5000 VersaProbe III. Ultraviolet-visible (UV-Vis) absorption spectra were recorded on an Agilent series UV-Vis-NIR spectrophotometer. Fluorescent quantum lifetimes were measured by fluorescence lifetime system (Light-Stone Instruments NTAS-TCSPC). Raman spectroscopy was performed on a HORIBA Scientific

LabRAM HR Raman spectrometer system. ICP mass spectrometry was performed on Agilent 730 series ICP-OES. Magnetic susceptibility was measured in a Quantum Design PPMS-9T. Low-temperature EPR spectra were recorded on a Bruker EMXplus.

## Electrochemical measurements

All Electrochemical characterizations were performed at room temperature (25 °C ± 2 °C) under dry air atmosphere on AutoLab PGSTAT302Nl with PINE AFMSRCE. The electrocatalyst was loaded on a rotating disk electrode (RDE) (disk area: 0.196 cm²) or a RRDE (disk area: 0.1866 cm², ring area: 0.2475 cm²) as working electrode. A graphite rod was used as counter electrode, Ag/AgCl electrode and 0.1 M KOH aqueous solution were used as reference electrode and electrolyte, respectively. First, 5 mg of the sample powder was mixed with 950 μL of ethanol and 50 μL of 5% wt% Nafion solution, then sonicated for 30 min to obtain a homogeneous catalyst ink. Next, 10 μL of the catalyst ink was deposited on a glassy carbon electrode and dried at room temperature (~25 °C), and the catalyst loading on glassy carbon electrode surface is ~50 μg (~0.25 mg cm⁻²) for RDE measurements. LSV curves were recorded at 225–2025 rpm with a sweep rate of 10 mV s⁻¹ using RDE between -0.95 and 0.05 V *vs.* Ag/AgCl. The potential measured in this study was aligned to the reversible hydrogen electrode (RHE) scale by using the Nernst equation from Ag/AgCl.

$$E_{RHE} = E_{Ag/AgCl} + 0.059pH + E^{\theta}_{Ag/AgCl} \qquad (6)$$

where $E_{Ag/AgCl}$ was the measured potential using Ag/AgCl, and $E^{\theta}_{Ag/AgCl}$ is the standard potential of Ag/AgCl of 0.1976 V.

The electron transfer number ($n$) was calculated from the Koutecky-Levich equation, which is expressed as follows:

$$\frac{1}{j} = \frac{1}{j_L} + \frac{1}{j_k} = \frac{1}{B\omega^{\frac{1}{2}}} + \frac{1}{j_k} \qquad (7)$$

where, $j$ is the measured current density, $j_k$ and $j_L$ are the kinetic-limiting and diffusion-limiting current densities, respectively, $\omega$ is the electrode rotating speed in rad s⁻¹. The K-L plots ($\omega^{-1/2}$ vs $j^{-1}$) in O₂-saturated 0.1 M KOH were determined from LSV curves at various rotation speeds. $B$ can be determined from the slope of Koutecky-Levich plots as given by

$$B = 0.62nFC_0(D_0)^{2/3}\upsilon^{-1/6} \qquad (8)$$

where $F$ is the Faraday constant ($F = 96485$ C mol⁻¹), $C_O$ is the bulk concentration of O₂ in 0.1 M KOH ($1.2 \times 10^{-6}$ mol cm⁻³), $D_O$ is the diffusion coefficient of O₂ in 0.1 M KOH ($1.9 \times 10^{-5}$ cm² s⁻¹ at 25 °C), and $\upsilon$ is the kinetic viscosity of 0.1 M KOH (0.01 cm² s⁻¹ at 25 °C).

The transferred electrons number ($n$) was determined by RRDE measurements. The HO₂⁻ (%) and the $n$ were calculated by the following equations:

$$(HO_2^-)\% = \frac{200 \times I_{Ring}}{I_{Ring} + I_{Disk} \times N} \qquad (9)$$

$$n = \frac{4 \times I_{Disk}}{I_{Disk} + I_{Ring}/N} \qquad (10)$$

where $I_{Disk}$ is the disk current, $I_{Ring}$ is the ring current, and $N$ (0.37) is the collection efficiency of RRDE.

The ECSA was determined by measuring the capacitive current associated with double-layer charging from the scan rate dependence of the CV. This measurement was performed on the same working electrode in a potential window of 1.00–1.10 V *vs.* RHE and scan rates ranging from 1.0 to 10.0 mV s⁻¹. Then linear fitting of the charging current density differences ($\Delta j = j_a - j_c$ at a potential of 1.05 V *vs.* RHE) against the scan rate was done. $j_a$ and $j_c$ represent the anodic and cathodic current densities, respectively. The slope is twice the double-layer capacitance $C_{dl}$. ECSA was calculated using ECSA=$R_f/m_{loading}$, where m$_{loading}$ is the loading mass of catalyst per geometrical area of the electrode. $R_f = C_{dl}/40$ μF cm⁻² (the average specific capacitance of a flat standard electrode with 1 cm² of real surface area is 40 μF cm⁻²). The EIS measurements were recorded in the frequency range from 100 kHz to 0.1 Hz with an AC signal amplitude of 10 mV in an N₂-saturated 0.1 M KOH electrolyte.

## Scanning electrochemical microscope

The SECM measurements were performed on a CHI920D, including a bi-potentiostat and a high-resolution 3D electrode positioner. The probe was a CHI116 10 μm Pt SECM tip. The substrate was a home-made concave Au electrode with a recessed microdisk with diameter of 25 μm at the depth of -7 μm. An Ag/AgCl electrode was used as reference electrode. A Pt wire was used as counter electrode. The electrolyte was Ar-saturated 0.1 M KOH aqueous solution with 0.5 mM FcMeOH.

The processing of the substrate was following the literature[51]. In brief, a 25 μm Au wire was inserted into a glass capillary tube (outer diameter of 2.2 mm), then heated with an oxyhydrogen flame to seal the Au wire into the tube. A copper wire was attached to one side of the Au wire by silver conductive paint. The end of the capillary containing the sealed Au wire is then polished flat with successively finer grit sandpaper (400-, 600-, 800-, 1000-, 3000-grit) until the exposure of Au surface. Then, the polished electrode was electrochemically etched in the CaCl₂ solution to dig a cavity. An optical microscope was used to check the sealing and the depth of the cavity.

To load catalysts into concave electrodes, catalyst powder was spread onto the surface of a glass slide. Then, the concave electrode was gently pressed onto the catalyst powder along vertical direction, and wiping the residuals with a lens wiping paper. An optical microscope was used to check the loading and flatness. The procedure may be repeated several times, until the cavity is filled with catalyst powder.

Probe approach curve technique was used to positioning $z$ axis and adjusting platform. The probe was biased at 0.5 V$_{Ag/AgCl}$ to oxidize FcMeOH to FcMeOH⁺. When the probe is approaching very close to the surface of substrate, the feedback current will be greatly change due to the restricted diffusion. A cutoff set at a current level of 75% to avoid crash. A quiet time was set for 20 s to steady current value. The adjustment of the platform was based on a three-point fix method.

SECM technique was used to locating the catalyst region. The probe was biased at 0.85 V$_{Ag/AgCl}$ and the substrate was biased at −0.9646 V$_{Ag/AgCl}$. Firstly, a 500 × 500 μm scan (5 μm/step) was performed for rough locating. Subsequently, a 200 × 200 μm scan (2 μm/step) was performed for fine positioning. The probe returned back to origin when the measurement finished. A quiet time was set for 20 s to steady current value. The TG/SC mode SECM images were recorded on this 200 × 200 μm region, but changing a series of the given substrate bias potential ranged from −0.9646 V$_{Ag/AgCl}$ to −0.0646 V$_{Ag/AgCl}$.

## Surface-interrogation (SI-) SECM measurement

The SI-SECM was performed on the same platform as SECM, including the instrument, electrodes and electrolyte.

Firstly, the $C_{dl}$ was measured by a series of CV scanning in a non-Faradaic potential window (0.05–0.15 V vs. RHE) at given scan rates of 10, 20, 30, 40, 50 mV s⁻¹. The ECSA$_{UME}$ (subscript UME is to declare the data obtained from ultra-micro electrode) was determined by the same procedure as described in the previous electrochemical measurements section.

Subsequently, the Pt tip was electrochemically cleaned by CV cycling from −0.9646 V$_{Ag/AgCl}$ to −0.4646 V$_{Ag/AgCl}$ at 50 mV s⁻¹ for -300 s. Then, the probe was switched to open circuit and the substrate

was subject to given bias potential from $-0.9646\,V_{Ag/AgCl}$ to $0.2354\,V_{Ag/AgCl}$ for 30 s. After that, the substrate was immediately switched to open circuit, while the probe was biased at $0.4\,V_{Ag/AgCl}$ for 30 s. The feedback current at probe ($I_{tip}$) was recorded. The electric quantity at tip ($Q_{tip}$) was derived by background subtraction and integration of $I_{tip}$. The active SD followed the relation as described by following equation:

$$SD = \frac{Q_{tip}/F \times N_A}{ECSA_{UME}} \qquad (11)$$

where $F$ represents the Faraday's constant ($96485\,C\,mol^{-1}$) and $N_A$ is the Avogadro constant ($6.02 \times 10^{23}\,mol^{-1}$).

The TOF was calculated by following equation:

$$TOF = \frac{N_A \times j_m}{SD_{mass} \times F} \qquad (12)$$

The kinetic mass activity ($j_m$) for the as-papered catalysts in this work is defined as:

$$j_m = \frac{j_k}{m_{catalyst}} \qquad (13)$$

where $m_{catalyst}$ is the catalyst loading on the glassy carbon disc ($mg\,cm^{-2}$), the potential to determine kinetic current density ($j_k$) is chosen at 0.8 V.

## XAFS measurements
X-ray absorption spectroscopy (XAS) experiments were carried out at the wiggler beamline BL10 at the DELTA storage ring (Dortmund, Germany) operated with 80–130 mA of 1.5 GeV electrons. Co K-edge spectra were collected using a Si (111)-channel cut monochromator and gas-filled ionization chambers as detectors for the incoming and the transmitted intensities, and a large area photodiode for the fluorescence photons.

## VF-SWV measurements
Square wave voltammetry tests were performed in $N_2$-saturated 0.1 M KOH electrolyte with a step potential of 10 mV, amplitude of 25 mV. The perturbation frequency ($f$) was varied from 1250 to 0.33 Hz.

## Electron transfer reaction between catalysts and TCNQ
5 mg of catalyst was exposed to 5 mL 7.65 mM TCNQ acetonitrile solution in a centrifuge tube through ultrasonic treatment, followed by stirring the suspension at 70 °C for 20 min to accelerate the electron transfer reaction. Subsequently, the supernatant is separated by centrifugation for further analysis by UV-Vis absorption spectra. The original TCNQ solution was diluted 250-fold with acetonitrile before measurement. The TCNQ solution reacted with the catalyst and was diluted twofold with acetonitrile before the measurements.

## Aqueous ZAB assembly and testing
An in-house-developed aqueous ZAB was built and assembled to evaluate the electrochemical energy conversion performance of the composite materials. It was constructed by pairing CoCOP-COOH@KB loaded onto a carbon paper (HCP020P, 0.19 mm thickness) with a Zn plate (0.20 mm thickness, 99.99% purity) in 6 M KOH (20 mL). The ZABs were assembled applying the following procedure: First, the air electrodes were prepared by pipetting catalyst slurry carefully onto the carbon paper, the catalyst loading on carbon paper is ~120 μg ($0.25\,mg\,cm^{-2}$). Subsequently, polished zinc plates served as the anode, zinc plates were manually polished with finer grit sandpaper (1000-, 2000-, 3000-grit) beforehand, followed by the sonication in deionized water for several minutes. Battery tests were carried out at room temperature (25 °C ± 2 °C) under a dry air atmosphere with a CHI

760E and a LAND CT2001A. The specific capacity and specific energy can be obtained from the consumed Zn after discharge.

## Data availability
The data that support the findings of this study are available from the corresponding author upon reasonable request.

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

## Acknowledgements

The authors acknowledge financial support from the National Natural Science Foundation of China (52373187, 52073137), the National Youth Top-notch Talent Support Program of China, the Natural Science Foundation of Jiangxi Province (20224ACB204006), the "Double Thousand Plan" Science and Technology Innovation High-End Talent Project of Jiangxi Province (jxsq2023201094), and the Open Research Fund (no. 2024JYBKF02) of Key Laboratory of Material Chemistry for Energy Conversion and Storage (HUST), Ministry of Education.

## Author contributions

B.H. performed the synthesis, structural characterizations, and electrochemical tests and analyzed the data. B.H., K.Y., and Y.C. wrote the paper. K.Y. and Y.C. supervised the entire project and were responsible for the infrastructure and project direction. D.L.-H. performed and analyzed the XAS data. Q.G. and X.T. helped to discuss the experimental data. All authors discussed the results, commented on them, and revised the manuscript.

## Competing interests

The authors declare no competing interests.
