## [Peer Review File · Nature Communications]

REVIEWER COMMENTS

Reviewer #1 (Remarks to the Author):

This research controlled the oxygen reduction activity of single atomic Co active sites using electron-withdrawing/donating functionality at the second shell. Additionally, a unique method was employed to measure the catalyst's site density and confirm the high intrinsic activity of Co single-atomic catalysts. The originality of this paper lies in systematically confirming the performance changes of the catalyst due to electron-withdrawing/donating effects and demonstrating reaction mechanisms and catalyst characteristics through various analyses. However, additional analysis seems necessary to explain the performance improvement. Therefore, this paper requires minor revision for publication in Nature Communications.

Reviewer #2 (Remarks to the Author):

In the manuscript., the authors reported a theoretical descriptor based on the binding energies of oxygen adsorbates, which is associated with the derived Sabatier volcano plot. The authors introduced several functional groups aimed at altering the binding energies of adsorbates. These functional groups work by withdrawing electrons from the metal center, thereby helping to lower the σ -bond. The authors validated this using both computational methods and XANES spectra.

The modulation of the microenvironment is a new area of study as scientists try to mimic the natural enzymes, where functional groups could significantly contribute to altering the local environment of the active site. For instance, they can influence local pH, the diffusion of reactants and products, proton transfer, and more. This manuscript solely concentrates on the impact of functional groups on the valence electrons of the metal center, which has been reported in several papers, such as ACS Catal. 12, 7278-7287, 2022. Utilizing binding energies as a descriptor is not a novel concept either (Nature Catalysis, 5, 615–623, 2022; J. Phys. Chem. Lett. 2018, 9, 588–595; Chem. Rev. 2018, 118, 2302–2312 ; Adv.Sci. 7, 1901614, 2020).

Some technical questions:

1. Why did the author omit the first step (adsorption of $*O_2$) in the calculation of the overpotential, as depicted in Fig. S3?
2. When introducing functional groups, there's a significant deformation observed in the COP structure. Did the author assess the stability of their models?

3. In Fig. 1C, the authors designate the rate-determining step as (RDS). However, in the text, only the potential-determining step (PDS) is mentioned.

4. Another crucial aspect is the author's determination of the rate-determining step (RDS). The desorption of $*OH \rightarrow * + OH^-$ is independent of the proton's activity. In contrast, the formation of $*OOH$, $*O$, $*OH$ depends on the proton concentration. These steps might be slower compared to the desorption of $*OH$, particularly when the proton activity is low in an alkaline environment. Consequently, determining the RDS solely on reaction energy could be misleading in this scenario.

5. Their Tafel slope is 41.1 (mV dec⁻¹). In a previous study (ACS Catal. 4, 4364–4376, 2014), it was indicated that a Tafel slope of 40 suggests 1~2 electrons transfer before the RDS. Consequently, the formation of $*O$ could potentially be the RDS, contradicting their theoretical suggestion that the desorption of $*OH$ is the RDS. Furthermore, their result show that there is a large change in the Tafel slope which suggest the change in the RDS. However, the calculation only shows the desorption of $*OH$ as the RDS.

Reviewer #3 (Remarks to the Author):

In this contribution, Huang et al. established the Sabatier plot by DFT calculation to forecast the catalytic O₂ reduction efficiency for a series of Co porphyrins with substitutions with different electron-withdrawing properties at the second coordination sphere of CoN₄ sites. It was shown that carboxyl group-substituted catalyst reduced the σ -bond orbital occupation, thus exhibiting an optimal OH* binding with Co center, which should be responsible for the best reactivity. The theoretical prediction was also well verified by rational design of different fringe groups modified Co porphyrins. Some related concerns are listed in the following:

(1) I do not agree with viewpoint of PDS on the $*OH$ desorption step for Co-N-C catalysts. Actually, the $*OH$ desorption step is usually deemed as RDS step for Fe-N-Cs rather Co-N-Cs due to the strong binding of OH intermediates on Fe sites. In contrast, the relatively weak oxophilicity for Co metal leads to the difficulty in the O₂ activation and readily OOH* desorption. As such, the GOOH for Co-N-C is supposed to located at weak binding side of Volcanic plot for 4e- pathway. For 2e- pathway, the OOH* desorption should be RDS, thereby locating at the strong adsorption side of volcanic plot. Theoretically speaking, the Co-N-Cs should be governed by 2e- pathway, which seems to contradict with conclusions described in the work, please, give a rational explanation

(2) The author should give the Co mass loading in the CoCOP-X@KB catalysts. Most importantly, considering that the 2e- + 2e- (O₂—H₂O₂; H₂O₂-H₂O) pathway is most likely occurring in the Co-based catalyst, the ORR activity test with different catalyst loading should be carried out to exclude the possible 2e- reduction reaction. Actually, the 2e- reduction pathway is readily covered when using enough high catalysts loading since generated H₂O₂ could be further reduced due to the long

residue time inside thick catalytic layer. I noticed that the H₂O₂ yield is high up to 20% in some catalysts in the RRDE test, which further strengthens the possibility of 2e⁻ reduction.

(3) Is it Co¹⁺ or Co²⁺ as active sites for O₂ activation? Please specify the Co electron valence for the COCOP-X@KB, and most importantly, the analysis of CV curves is necessary for the clarity of catalytic active sites for this series of catalysts.

(4) The limited diffusion current shown in the ORR measurement is low (ca. 4 mA/cm²), please try to give a reasonable explanation.

Manuscript NCOMMS-23-53238-point-by-point responses to the reviewer's comments

Reviewer #1:

This research controlled the oxygen reduction activity of single atomic Co active sites using electron-withdrawing/donating functionality at the second shell. Additionally, a unique method was employed to measure the catalyst's site density and confirm the high intrinsic activity of Co single-atomic catalysts. The originality of this paper lies in systematically confirming the performance changes of the catalyst due to electron-withdrawing/donating effects and demonstrating reaction mechanisms and catalyst characteristics through various analyses. However, additional analysis seems necessary to explain the performance improvement. Therefore, this paper requires minor revision for publication in Nature Communications.

Response: We appreciate your very positive review and recommendation that our manuscript could be accepted for publication in *Nature Communications* after minor revision. The answers to the questions you raised are addressed in detail as below.

1. Among the reported literatures (J. Am. Chem. Soc. 2019, 141, 31, 12372–12381, Adv. Mater. 2023, 35, 2210550), it is known that Fe has a strong binding to oxygen intermediates, whereas Co is known to have weak oxygen intermediates adsorption binding energy. Single atomic Fe sites can control the decreasing adsorption of oxygen intermediates and ORR activity through the electron-withdrawing effect. This paper demonstrates through calculations and experiments that the performance was enhanced by the withdrawing effect of functionality in Co active site's second shell. What are the differences between the reported papers and the calculations/experiments showcased in this paper?

Response: Thanks for your feedback. As is known, for ORR the last hydrogenation step ($*\text{OH} \rightarrow \text{OH}^-$) appears to be the potential-determining step (PDS) for nearly all the atomically dispersed transition metal-nitrogen-carbon (M-N-C) catalysts due to the hybridization of metal d and O p orbitals. For M-N-C catalysts, the N atoms not only act as the anchoring sites to stabilize the single metal atoms but also play important roles in modulating the electronic structures of the active sites. Thereinto, the strong electronegativity of N atom can undesirably alter the electronic properties of metal catalytic center and increase the free adsorption energy of oxygen reaction intermediates (please refer to *J. Phys. Chem. C* **2020**, 124, 13168–13176; *Adv. Energy Mater.* **2020**, 10, 2002896; *J. Am. Chem. Soc.* **2019**, 141, 20118–20126). Among transition metal-based single-atom catalysts, only Ni atom is universally recognized as the active center with relatively weak oxygen binding affinity, while the other transition metals all exhibit different oxygen affinity in their corresponding electronic configuration microenvironment (please refer to *Nat. Commun.* **2023**, 14, 1792; *Adv. Funct. Mater.* **2023**, 33, 2210867).

On the basis of the Sabatier principle, the appropriate binding energy of oxygen adsorbates on metal center can lower the kinetic energy barrier and benefit the catalytic activity. Thereinto, the catalytic theoretical overpotential is largely

microenvironment-dependent thus the local microenvironment of metals should be precisely controlled. In addition, the topological configuration of the overall polymer skeletons surrounding the active sites also affects its oxygen affinity. In the offered literature (*J. Am. Chem. Soc.* **2019**, 141, 31, 12372–12381), the authors investigated the effect of the nature of 3d metal within a series of M–N–C catalysts on the electrocatalytic activity/selectivity for ORR. Differently, their M–N–C materials were prepared through the pyrolysis of well-managed precursors, thus the pyrolysis process probably triggers internal structural collapse or even reconstruct the precast well-managed precursor structure, resulting in the formation of random and complex sites and different adsorption behaviors of Co-N₄ centers. In the offered literature (*Adv. Mater.* **2023**, 35, 2210550), the authors prepared an air cathode catalyst by finely tuning the fluorinated nanopores of a COF. However, for polymer-based catalysts, the locally carbon structure surrounding the metal center and the overall topological skeletons of polymer play crucial role in intrinsic catalytic activity. Also, the π -conjugated electrons in the connected carbon matrix and the electronic configuration of the carbon atom covalent-connected to the Co-N₄ sites both determine the adsorption behavior. Normally, the conjugated polymers with larger intralayer delocalized π -electron systems tend to relatively stronger binding strength with oxygen adsorbates since the strong hybridization between *d* orbitals of π -conjugated polymers and oxygen *p* orbitals enhance intermediate adsorption (please refer to *Nat. Commun.* **2022**, 13, 57; *Front. Mater.*, **2019**, 6, 244). Meantime, the design of supported nanocomposite metal catalysts is of great importance for maximizing catalytic activity and the interaction between metal species and the support strongly impacts the ORR performance. The metal-support interaction induces the charge transfer from/to metal centers, and graphene-like support can modify the electronic structure of metal by virtue of the formation of electronic channels via π - π conjugation of carbon support and porphyrin-based CoCOP-X polymers. Therefore, unlike the literature (*Adv. Mater.* **2023**, 35, 2210550) where LDH substrate merely act as hydrophilic surface, the Ketjen black serves as a conductive support in our work thus the obtained supported nanocomposite CoCOP-X@KB catalysts exhibit different adsorption behaviors.

The Co-N₄ configuration has been extensively explored due to its positioning at the top of volcano plot for the ORR limiting potential against the oxygen adsorption free energy. Further, through various optimization strategies, the intrinsic performance of Co-N₄ site has been enhanced. Among the literatures, many studies reported that the original Co-N₄ sites require modification to reduce the thermodynamic barrier for OH* desorption for further final product release, thus increasing the catalytic efficiency of catalysts (please refer to *J. Am. Chem. Soc.* **2019**, 141, 20118–20126; *J. Am. Chem. Soc.* **2017**, 139, 17269–17272; *Adv. Mater.* **2022**, 34, 2204021; *Adv. Mater.* **2019**, 31, 1901666; *Adv. Energy Mater.* **2020**, 10, 2002896; *Adv. Funct. Mater.* **2023**, 33, 2210867; *Chem. Eng. J.* **2022**, 430, 132642; *Chem. Eng. J.* **2022**, 429, 132119). Meanwhile, it is well-known that the binding of oxygen species at the porphyrin induces the charge transfer towards the oxygen intermediates, therefore the negatively charged and nucleophilic characters of the oxygen make it tightly bind to the

positively charged active center.

In our work, the theoretical Sabatier volcano plot was primarily derived and forecasted the tough *OH release on cobalt porphyrins and the incorporation of electron-withdrawing substituents on porphyrin secondary coordination sphere is deemed to alleviate the overmuch *OH binding strength. On this basis, the secondary sphere microenvironment customization strategy was proposed to promote the catalytic efficiency of cobalt porphyrin-based polymers and experimentally validate the Sabatier principle. Thereinto, the carboxyl-substituted porphyrin model with an electron-deficient Co center is expected to approach the top of the Sabatier volcano map and emerges the highest ORR reactivity. From the experimental perspective, systematic characterizations have been applied to demonstrate the oxygen affinity of catalysts and track the dynamic evolution between metal center and oxygen intermediate for elucidating the catalytic mechanism.

As shown in revised **Fig. 6**, progressive testing techniques incorporating electrochemical measurements and *in situ* characterization approaches were firstly used to achieve a real-time analysis for ascertaining the intermediates absorbed at Co sites. The ATR-SEIRAS manifests the consumption of dissolved oxygen (**Fig. 6a, b**). The *in situ* electrochemical Raman spectra under a series of applied bias voltages on CoCOP-COOH@KB (**Fig. 6c**) confirms that the *OH species acts as the key intermediate and the final *OH desorption step is the PDS for ORR on CoCOP-COOH@KB catalyst, which is in good agreement with DFT calculations.

Additionally, we supplemented the zero-field cooling temperature-dependent magnetic susceptibility measurements and electron paramagnetic resonance (EPR) spectra of CoCOP-COOH@KB, CoCOP-H@KB and CoCOP-CH₃@KB to verify the spin state of their Co centers. The average number of the unpaired Co 3d electrons acquired from the magnetic susceptibility plots (**Fig. 6d**) decreases along with the electron-withdrawing property of secondary sphere substituent increases. This result suggests that the strong electron-withdrawing/donating effect induces the Co 3d electronic spin state transition, and the Co 3d shells possesses different electronic configurations, which are high spin state with $d_{xy}2d_{xz}2d_{yz}1d_z^21d_x^2-y^21$ and medium spin state with $d_{xy}2d_{xz}2d_{yz}2d_z^21$ (**Fig. 6e**). Further, the shape and intensity of EPR profile are impacted by the lone pair electrons onto the structural transition (**Fig. 6f**). The EPR spectrum shows an obvious signal at $g \approx 2.00$ for CoCOP-CH₃@KB and CoCOP-H@KB, which is attributed to a typical high spin Co²⁺ signal. As the enhancement of the electron-withdrawing property on substituent, the spin structure of catalytic Co sites changes from high spin (CoCOP-CH₃@KB and CoCOP-H@KB) to medium spin (CoCOP-COOH@KB). The EPR results consequently indicates the weakened O₂ adsorption affinity of CoCOP-COOH@KB, which is in agreement with the theoretical calculation. Thermodynamically, the relatively weak adsorption of *OH permits the rapid final *OH release, and the $d_x^2-y^2$ and d_{xy} orbitals are neglected due to their negligible interactions with oxygen intermediates. Compared with high spin Co centers within CoCOP-CH₃@KB, the decreased proportion of electrons in bonding orbital of medium spin Co²⁺ within CoCOP-COOH@KB leads to an alleviative Co/*OH interaction (**Fig.6g**), which is more favorable for propelling ORR

cycles.

Thus, based on above results, it can be concluded that using both computational and experimental methods, the secondary sphere microenvironment customization strategy establishes the appropriate Co-oxygen binding to promote the catalytic efficiency of cobalt porphyrin-based polymers and experimentally validate the Sabatier principle. Thereinto, the incorporation of electron-withdrawing carboxyl substituents is deemed to alleviate the *OH binding strength and the electron-deficient Co center is expected to approach the top of the Sabatier volcano map and emerges the highest ORR reactivity.

The added discussion about Revised Fig. 6:

In the meantime, the electron spin configuration and unpaired electrons on Co ions within CoCOP-COOH@KB, CoCOP-H@KB and CoCOP-CH₃@KB counterparts were probed by the zero-field cooling temperature-dependent magnetic susceptibility and ex situ cryo (100 K) electron paramagnetic resonance (EPR) spectroscopy. The average number of the unpaired Co 3d electrons acquired from the magnetic susceptibility plots (**Fig. 6d**) decreases along with the electron-withdrawing property of secondary sphere substituent increases. This result suggests that the strong electron-withdrawing/donating effect induces the Co 3d electronic spin state transition, and the Co 3d shells possesses different electronic configurations, which are high spin state with $d_{xy}2d_{xz}2d_{yz}1d_z^21d_x^2-y^21$ and medium spin state with $d_{xy}2d_{xz}2d_{yz}2d_z^21$ (**Fig. 6e**). Further, the shape and intensity of EPR profile are impacted by the lone pair electrons onto the structural transition (**Fig. 6f**). The EPR spectrum shows an obvious signal at $g \approx 2.00$ for CoCOP-CH₃@KB and CoCOP-H@KB, which is attributed to a typical high spin Co²⁺ signal. As the enhancement of the electron-withdrawing property on substituent, the spin structure of catalytic Co sites changes from high spin (CoCOP-CH₃@KB and CoCOP-H@KB) to medium spin (CoCOP-COOH@KB). The EPR results consequently indicates the weakened O₂ adsorption affinity of CoCOP-COOH@KB, which is in agreement with the theoretical calculation. Thermodynamically, the relatively weak adsorption of *OH permits the rapid final *OH release, and the $d_x^2-y^2$ and d_{xy} orbitals are neglected due to their negligible interactions with oxygen intermediates. Compared with high spin Co centers within CoCOP-CH₃@KB, the decreased proportion of electrons in bonding orbital of medium spin Co²⁺ within CoCOP-COOH@KB leads to an alleviative Co/*OH interaction (**Fig.6g**), which is more favorable for propelling ORR cycles.

Also, we appreciate the reviewer for providing us the literatures, we have appropriately cited these literatures in our revised manuscript as references 11 and 27 for better claiming the relation between the oxygen adsorption and the ORR activity of catalysts.

The added references:

11. Sun, Y. et al. Activity-selectivity trends in the electrochemical production of

hydrogen peroxide over single-site metal-nitrogen-carbon catalysts. *J. Am. Chem. Soc.* **141**, 12372-12381 (2019).

27. Cao, Q. et al. A fluorinated covalent organic framework with accelerated oxygen transfer nanochannels for high-performance zinc-air batteries. *Adv. Mater.* **35**, 2210550 (2023).

Fig. 6 | The *in situ* electrochemical spectroscopic characterizations and schematic diagram of potential determining step transition. (a) Operando ATR-SEIRAS spectra for ORR at 500-2000 cm^{-1} , (b) potential dependent O-O stretching peak intensity, (c) potential-dependent *in situ* Raman spectra during ORR process over CoCOP-COOH@KB catalysts in 0.1 M KOH solution. (d) Magnetic susceptibility of CoCOP-COOH@KB, CoCOP-H@KB and CoCOP-CH₃@KB catalysts. M.S.

represents medium spin, H.S. represents high spin. (e) 3d electron occupations in the M.S. and H.S. states of Co²⁺ in CoCOP-X@KB. (f) Low-temperature EPR spectra recorded at T = 100 K of CoCOP-COOH@KB, CoCOP-H@KB and CoCOP-CH₃@KB catalysts. (g) The orbital interactions between Co centers with different spin states and the potential determining *OH intermediates.

2. In the calculations, COCH₃ was found to have lower performance compared to pristine. The authors claimed that the performance improved due to the electron-withdrawing effect. COCH₃ also exhibits an electron-withdrawing effect, and experimental results in Figure 4 show higher performance than pristine. The authors need to explain the reason for the discrepancy between the experimental results and the calculations.

Response: Thanks for your feedback. In our work, DFT calculations are primarily applied to screen the effect of electron withdrawing/donating substituent groups on porphyrin based Co-N₄ sites for qualitatively the performance optimization trends. This obtained Sabatier volcano plot instructs that electron-withdrawing substituents mitigate the over-strong *OH intermediate adsorption by virtue of the decreased proportion of electrons in bonding orbital. In our work, Por-COCH₃ model was found to have lower performance compared to pristine Por-H model, we attribute this phenomenon to the generally inevitable calculational uncertainty at the applied level of density functional theory (please refer to *J. Phys. Chem. C* **2016**, 120, 24910–24916; *ACS Catal.* **2016**, 6, 8, 5251–5259; *Phys. Rev. D* **2016**, 93, 114018). The uncertainty of DFT calculations within the commonly accepted energy ranges of chemical accuracy may cause the deviation between theoretically predicted and experimental results. During DFT calculation, by identifying and rectifying the systematic error, the calculational uncertainty can be significantly reduced and scaling relation is sufficiently accurate to be applicable in catalysis research. In our work, the theoretical predictions are also detailly discussed to guide the design and fabrication of Co porphyrin-based polymer nanocomposite analogs catalysts. Finally, our finding not only highlights the exploitation on electronic microenvironment customization towards enhanced activity of atomic metal, but also sharpens the understanding on designing efficient M-N-C catalysts with the aid of experimentally validated Sabatier plot.

3. In figure 4, the performance of commercial Pt/C in the ORR half-cell test seems necessary. Furthermore, it was claimed that catalyst have better durability than commercial Pt/C in the half-cell test. It is requested to present data on the actual full cell zinc-air battery durability.

Response: Thanks for your constructive suggestion. We agree that the full cell zinc-air battery durability is important for verifying the superior stability of our well-designed CoCOP-COOH@KB catalyst to commercial Pt/C. Hence their rechargeability were then tested (Supplementary Fig. 46c) and the corresponding discussion have been added in our revised manuscript.

The added discussion about Revised Supplementary Fig. 46c:

The rechargeability and cyclic durability of CoCOP-COOH@KB and Pt/C counterpart were then tested (Supplementary Fig. 46c). Note that the CoCOP-COOH@KB catalyst demonstrates superior cycling stability and can be stably charged and discharged over 300 hours at a current density of 25 mA cm^{-2} . After long-term operation, the negligible voltage-gap increase of CoCOP-COOH@KB-based ZAB indicates its excellent durability, which outperforms that of the commercial Pt/C catalysts.

Revised supplementary Fig. 46. (c) Galvanostatic cycling stability at 25 mA cm^{-2} of ZABs using CoCOP-COOH@KB and Pt/C as cathode catalysts.

4. The Site density (SD) and turnover frequency (TOF) measurement method appear interesting and were reasonably introduced in the authors' previous paper. However, while this paper claims a high TOF for Co active sites, the actual performance in ORR half-cell test seems lower when compared to the literature presented in table S2. Although there might be differences in measurement methods, it seems necessary to demonstrate whether the high TOF is observed even when measured using the same method for comparison with other papers.

Response: Thanks for your very constructive suggestion. The reviewer mentioned that the performance of CoCOP-COOH@KB in ORR half-cell test seems lower when compared to the literature presented in Supplementary Table 2 (Supplementary Table 3 in revised supplementary information). We have checked the specific performance in ORR half-cell test of the corresponding catalysts mentioned in the literatures and found that their half-wave potentials ($E_{1/2}$) are basically 100 mV lower than CoCOP-COOH@KB, thus exhibiting lower TOF values. The TOF and kinetic mass activity (j_m) values were calculated by following equations:

$$\text{TOF} = \frac{N_A \times j_m}{\text{SD}_{\text{mass}} \times F}$$

$$j_m = \frac{j_k}{m_{\text{catalyst}}}$$

where F represents the Faraday's constant (96485 C mol^{-1}), N_A is the Avogadro constant ($6.02 \times 10^{23} \text{ mol}^{-1}$), SD_{mass} is the number of sites per catalyst mass, m_{catalyst} is the catalyst loading on the glassy carbon disc (mg cm^{-2}) and j_k is kinetic current density.

For FeNC-800 catalyst (*ACS Catal.* **2019**, 9, 4841–4852), it exhibits a similar $E_{1/2}$ of approximately 0.88V, however the lower TOF values of FeNC-800 was largely attributed to its sampling under the low applied bias voltage. Also, we should emphasize that, as shown in the above equations, the TOF value is directly proportional to kinetic current density (j_k , which is dependent on the applied bias voltages). Generally, when the j_k under more negative applied bias voltages is chosen for calculating TOF, the TOF value would exhibit sharp increase. In our work, we also performed the TOF value calculation based on the potential of 0.8 V which is in the vast majority of the mentioned literatures, hence it is the reason why our TOF value is significantly superior to that of FeNC-800 catalyst. This is the main reason that the CoCOP-COOH@KB claims a higher TOF for Co active sites than FeNC-800 catalyst in literature (*ACS Catal.* **2019**, 9, 4841–4852).

In addition, it is reported that there are indeed certain degrees of variation in the active site density obtained from different measurement methods. The common measurement methods generally involve CO adsorption, nitrite reduction, thereinto CO adsorption overestimated site density while nitrite reduction underestimated it (please refer to *Nature Catalysis* **2022**, 5, 163–170). In contrast, scanning electrochemical microscopy has emerged as a powerful tool in studying the behavior of ensemble active sites that **truly participate in the electrocatalytic reaction** (please refer to *Nature Catalysis* **2021**, 4, 615–622; *ACS Energy Lett.* **2019**, 4, 1793–1802; *Angew. Chem. Int. Ed.* **2020**, 59, 16376–16380; *Acc. Chem. Res.* **2022**, 55, 5, 759–769).

In our work, unlike CO adsorption method which explores the physically meaningful SD and TOF values, the SECM technique provides a straightforward approach to quantify the **electrochemically accessible active sites that truly participate in ORR** in real time. Therefore, SECM technique is a versatile method for obtaining SD and TOF in electrocatalytic reaction, as such, it is believed that our TOF values are scientific.

5. The authors should demonstrate the uniform coating thickness of the polymer on KB rather than SEM images. The difference in thickness can indicate differences in exposed sites and may also affect mass transport by blocking pores.

Response: Thanks very much for your feedback. In the manuscript, systematic electron microscopic techniques such as high-resolution TEM and aberration-corrected HADDF-STEM clearly visualize the homogeneous recommendation between the CoCOP-X polymer with the carbon black conductive support. The slightly dark regions in electron microscopic images (Fig. R1-R4) are attributed to the presence of vertically aligned carbon black materials with spherical morphology, the distribution of the CoCOP-X@KB is highly uniform (please refer to *Adv. Energy Mater.* **2023**, 13, 2300325; *Appl. Catal. B-Environ.* **2022**, 315, 121590).

The porous characteristics of the KB support and CoCOP-X@KB materials were assessed by nitrogen adsorption/desorption measurements (Supplementary Fig. 26, 27), and provided auxiliary evidence for the uniform distribution throughout CoCOP-X@KB. Notably, KB and CoCOP-X@KB samples all showed type IV isotherms and

H3 hysteresis loops, thus the almost unchanged mesopore structure of CoCOP-X@KB series along with the substituent alternation further confirms the uniform coating of CoCOP-X polymer on KB. Moreover, the well-maintained skeletons and similar rich mesopores of CoCOP-X@KB catalysts favor the full exposure of active sites and provide more mass transport channels for electrocatalytic oxygen reduction.

From the perspective of catalysts' preparation, CoCOP-X@KB were fabricated by compositing the as-prepared CoCOP-X polymers with carbon black through ball milling. During the ball milling process, CoCOP-X polymers were exfoliated into structurally well-defined structures while being loaded onto the surface of carbon black. As we known, the electrocatalytic activity of KB towards ORR is very limited (please refer to *Adv. Energy Mater.* **2023**, 13, 2300325; *Angew. Chem. Int. Ed.* **2021**, 60, 8472-8476), thereinto the KB is mainly served as the conductive substrate. The CoCOP-X polymer, as the truly active center, is uniformly loaded on the outer surface of KB with similar thickness.

Additionally, compared to carbonaceous materials, the pyrolysis-free materials with relatively poor electrical conductivity further affects the accurate evaluation on ORR reactivity. Thereby in many previously reported studies (please refer to *Nat. Commun.* **2022**, 13, 57; *Angew. Chem. Int. Ed.* **2021**, 60, 8472-8476; *Adv. Mater.* **2022**, 34, 2204570; *ACS Catal.* **2022**, 12, 3138–3148; *Nano Res.* **2022**, 15, 1942–1948; *ACS Energy Lett.* **2019**, 4, 2251–2258), when preparing catalyst ink for half-cell test, the corresponding pyrolysis-free materials are normally mixed with carbon black or graphene to improve its conductivity and the dispersion on the glassy carbon electrode, without considering the influence of carbon black. Therefore, these all provide solid proof that the mixture between KB and CoCOP-X polymers is homogeneous and the thickness of composite material is uniform, the addition of KB does not significantly affect the accessible active sites.

Fig. R1. The TEM image of CoCOP-COOH@KB.

Fig. R2. The TEM image of CoCOP-CH₃@KB.

Fig. R3. The TEM image of CoCOP-H@KB.

Fig. R4. The HADDF-STEM image of CoCOP-H@KB.

Supplementary Fig. 26. N₂ adsorption-desorption isotherms of CoCOP-CH₃@KB, CoCOP-H@KB, CoCOP-COCH₃@KB, CoCOP-COOCH₃@KB, CoCOP-COOH@KB, CoCOP-CN@KB and KB.

Supplementary Fig. 27. Pore size distribution profiles of CoCOP-CH₃@KB, CoCOP-H@KB, CoCOP-COCH₃@KB, CoCOP-COOCH₃@KB, CoCOP-COOH@KB, CoCOP-CN@KB and KB.

6. It seems that the BET surface area can differ by up to 100 m²/g in table S1. To discuss the withdrawing/donating effects of Co active sites, the authors need to demonstrate if the support structure is truly identical in the synthesis process. Furthermore, due to the influence on electronic structure based on site density, it would be good to provide ICP data on site density for all samples to ensure they have the same site density.

Response: Thanks very much for your suggestion. Primarily, the surface area of materials can be influenced by the steric bulkiness of functional substituent on secondary coordination sphere of porphyrin monomers. The introduction of substituents will alter the skeleton configuration of polymer, and the pores and channel of polymers may be either blocked or expanded depending on the substituents molecular volume. According to previously reported studies (please refer to *Chem. Mater.* **2016**, 28, 1489–1494; *Adv. Mater.* **2018**, 30, 1706330; *Chem. Eur. J.* **2010**, 16, 1137–114; *J. Energy Chem.* **2022**, 65, 490–496; *J. Am. Chem. Soc.* **2009**, 131, 17, 6111–6113), compared to the original polymer with undecorated pores, the substituents with relatively small size on secondary coordination sphere can partly block the polymer channels owing to its steric bulkiness, thus resulting in decreased specific surface area. In contrast, there is a dependence of the longitudinal interlayer spacing on the secondary sphere substituent configuration, substituents with the molecular size larger than the original undecorated polymer pores may cause slight topological in-plane architecture deformation (please refer to *J. Am. Chem. Soc.* **2020**, 142, 17524–17530). As a result, in comparison to the polymers modified with small-sized substituents, large-sized substituents permit a slightly larger surface area. Hence, the need for topological control of polymer skeletons must be balanced with the need to create maximum porosity. Thereinto, the suitable-sized carboxyl substituent strikes the desired balance between the in-plane architecture deformation and pore channels blockage, thus achieving the largest surface area and superior intrinsic ORR reactivity.

In the meantime, the significant changes in surface area caused by the alteration of substituent size with the differences of even up to 100 m²/g have been regularly reported (please refer to *J. Am. Chem. Soc.* **2020**, 142, 17524–17530; *Adv. Mater.* **2022**, 34, 2204570; *Angew. Chem. Int. Ed.* **2023**, 62, e202304608; *Angew. Chem. Int. Ed.* **2024**, 63, e202314988). In summary, we firmly believe that the supporting structures for all catalysts are all identical and they have not impacted the modulation on Co center intrinsic activity cause by secondary sphere microenvironment customization strategy.

We also agree that ICP data on site density for all samples should be provided to ensure the same site density. Thereby the corresponding ICP data have been added and discussed in our revised manuscript (please see Supplementary Table 1).

The added discussion: The Co contents, measured by inductively coupled plasma optical emission spectroscopy (ICP-OES), were similar in all the catalysts, which guaranteed that the effect of molecular level microenvironment customization on ORR reactivity of CoCOP-X@KB catalysts can be explored independently

(Supplementary Table 1).

Supplementary Table 1. Mass contents of the metallic elements in CoCOP-X@KB.

Sample	Co contents (wt%)
CoCOP-COOH@KB	0.17
CoCOP-CH ₃ @KB	0.15
CoCOP-H@KB	0.14
CoCOP-COCH ₃ @KB	0.13
CoCOP-COOCH ₃ @KB	0.14
CoCOP-CN@KB	0.19

7. In figure 3b, the authors presented the change in oxidation states using XANES. However, the range shown by the authors is situated on the oxidized side for the Co foil. It would be better to show the trend for Co foil, oxide, and samples in an appropriate range. Since there is no heat treatment process, it is believed that there is no aggregated Co and that it would have the same atomic structure. However, it is necessary to demonstrate through fitting data whether there are structural changes in the Co-N₄ site depending on the change in the second shell.

Response: Thanks for your very constructive suggestion. We have carefully checked and confirmed a more suitable range of normalized Co K-edge XANES spectra for better claiming the oxidation states change for CoCOP-X@KB catalysts. Obviously, compared with Co foil, CoO control samples and CoCOP-H@KB counterpart, the enhanced absorption edge at around 7720 eV of CoCOP-COOH@KB indicates its higher oxidation state of the X-ray absorbing Co center, which originates from the electron transfer from the Co-N₄ site to the carboxyl substituent group.

Revised Fig. 3b. Normalized Co K-edge XANES spectra (up), and the spectra at the pre-edge region (bottom left), white-line rising edge region (bottom right) of CoCOP-COOH@KB, CoCOP-H@KB and CoCOP-CH₃@KB in comparison to a Co metal foil, CoO, and CoTPP serving as control samples.

Also, we added the EXAFS fitting analysis for CoCOP-COOH@KB, CoCOP-H@KB and CoCOP-CH₃@KB, the EXAFS fitting shows that their isolated Co were all coordinated with about 4 nitrogen atoms on average, confirming that the existence of Co-N₄ configuration. The minor peak at around 2.4 Å is corresponding to the second paths of long-range Co-C and Co-N coordination (please refer to *Nat. Commun.* **2023**, 14, 172; *Sci. Adv.* **2020**, 6, eaaz4824; *Nat. Commun.* **2013**, 4, 2076). Overall, the results above reveal the similar coordination environment of Co center and minimal structural changes in the Co-N₄ configurations of CoCOP-X@KB series catalysts.

Fig. R5. EXAFS fitting curves in the R-space of (a) CoCOP-COOH@KB, (b) CoCOP-H@KB and (c) CoCOP-CH₃@KB.

Table R1. EXAFS fitting parameters at the Co K-Edge for CoCOP-COOH@KB, CoCOP-H@KB and CoCOP-CH₃@KB.

Sample	Shell	CN	R(Å)	$\Delta\sigma^2 \cdot 10^3$ (Å ²)	r-factor
CoCOP-COOH@KB	Co-N	4.0	1.92	0.7	0.38
	Co-C	6.6	2.3	2.4	
CoCOP-H@KB	Co-N	4.1	1.92	1.54	0.24
	Co-C	7.1	2.4	7.4	
CoCOP-CH₃@KB	Co-N	4.2	1.93	5.6	0.03
	Co-C	6.8	2.89	3.3	

Reviewer #2:

In the manuscript., the authors reported a theoretical descriptor based on the binding energies of oxygen adsorbates, which is associated with the derived Sabatier volcano plot. The authors introduced several functional groups aimed at altering the binding energies of adsorbates. These functional groups work by withdrawing electrons from the metal center, thereby helping to lower the σ -bond. The authors validated this using both computational methods and XANES spectra.

The modulation of the microenvironment is a new area of study as scientists try to mimic the natural enzymes, where functional groups could significantly contribute to altering the local environment of the active site. For instance, they can influence local pH, the diffusion of reactants and products, proton transfer, and more. This manuscript solely concentrates on the impact of functional groups on the valence electrons of the metal center, which has been reported in several papers, such as ACS Catal. 12, 7278-7287, 2022. Utilizing binding energies as a descriptor is not a novel concept either (Nature Catalysis, 5, 615–623, 2022; J. Phys. Chem. Lett. 2018, 9, 588–595; Chem. Rev. 2018, 118, 2302–2312; Adv.Sci. 7, 1901614, 2020).

Response: We sincerely thank the reviewer for providing constructive feedbacks to improve the quality of our manuscript. We have revised the manuscript according to your valuable comments, and supplied detailed discussion.

We appreciate the reviewer for providing us the literatures, after carefully reading through these literatures, we have identified many significant differences between our work and literature works. In the following section, the differences are discussed in detail:

1) ACS Catal. 12, 7278-7287, 2022

First, in the mentioned research, the authors prepared a series of electronically tunable Fe centers through adjusting electron-withdrawing/donating substituents of phthalocyanine. They showed the effect of functional groups alternation on the ORR activity via electrochemical measurements. However, they did not provide an elucidation of the underlying mechanisms for the electrocatalytic performance enhancement. In contrast, we systematically validated structure-reactivity relations from both theoretical and experimental perspectives. First, we theoretically present a descriptor based on the correlation between Co-N₄ ORR overpotential and oxygen adsorption behaviors on Co porphyrin model, and put forward a descriptor-derived Sabatier-type volcano map to achieve reactivity prediction. Then, we prepared a series of cobalt porphyrin-based polymer nanocomposite analogs and realized the alternation of chemical microenvironmental properties on Co center by secondary sphere microenvironment customization strategy. Systematic X-ray spectroscopic and *in situ* electrochemical characterizations were further affirmed the oxygen intermediates adsorption behaviors and dynamic evolution.

More importantly, the functionalized Fe-phthalocyanine-based catalysts in this mentioned research were obtained by partially replacing the bridging pyromellitic diimide with terminal phthalimide bearing different functional groups thus anchoring the functional groups on the edge of the polymer. As a result, their regulation is

primarily effective at the peripheral metal centers, whereas the modulation on catalytic sites within the overall polymer is limited. On the contrary, porphyrin enables systematical modifications at *meso*-, β -positions and the secondary coordination sphere, while persisting original geometries and coordination characteristics nearly invariable. By virtue of the incorporation of substituents into each porphyrin monomer, the homogeneous electronically well-manipulated active sites can be fabricated, which can be conducted to better probe the accurate structure-property-activity relations on M-N-C catalysts. To sum up, our work is completely different from this literature.

2) Nature Catalysis, 5, 615–623, 2022; J. Phys. Chem. Lett. 2018, 9, 588–595; Chem. Rev. 2018, 118, 2302–2312; Adv.Sci. 7, 1901614, 2020

In addition to the literatures mentioned by the reviewer, the binding energy is found to be a good descriptor for the ORR performance prediction. After carefully reading the mentioned works, we found that the majority of the structure-performance relations in these researches are based on Density Functional Theory (DFT) calculations, without the experimental validation. In the provided literatures, only very few research articles have experimentally verified the binding energy descriptor and reactivity volcano, but they are all based on the active centers of noble metals or metal oxides-based materials. Hence, it is urgently needed to establish apparent oxygen binding energy as the descriptor based on atomically dispersed M-N-C catalysts for foretelling the ORR activity trend of efficient non-noble catalysts. To solve this issue, our prepared pyrolysis-free-based materials are acted as an ideal platform to get insight into the structure-activity relationships during ORR process due to their well-defined structure benefitting from the mild synthesis route. This work offers ample strategies for designing single-atom catalysts with well-managed microenvironment under the guidance of Sabatier volcano map, so we believe that our work is completely different from these literatures.

Also, we appreciate the reviewer for providing us the literatures, we have appropriately cited these literatures in our revised manuscript as references 28, 36, 38 and 39 for better verifying the importance of binding interaction as the descriptor for predicting ORR activity.

The added references:

28. Yuan, S. et al. Tuning the catalytic activity of Fe-phthalocyanine-based catalysts for the oxygen reduction reaction by ligand functionalization. *ACS Catal.* **12**, 7278-7287 (2022).

36. Luo, M. & Koper, M. T. M. A kinetic descriptor for the electrolyte effect on the oxygen reduction kinetics on Pt(111). *Nat. Catal.* **5**, 615-623 (2022).

38. Krishnamurthy, D., Sumaria, V. & Viswanathan, V. Maximal predictability approach for identifying the right descriptors for electrocatalytic reactions. *J. Phys.*

Chem. Lett. **9**, 588-595 (2018).

39. Liu, J. et al. Progress and challenges toward the rational design of oxygen electrocatalysts based on a descriptor approach. *Adv. Sci.* **7**, 1901614 (2020).

Some technical questions:

1. Why did the author omit the first step (adsorption of *O₂) in the calculation of the overpotential, as depicted in Fig. S3?

Response: Thanks for your feedback. It is well-known that the ORR pathway energetics is usually determined by the free energy and binding strength of three reaction intermediates (*OOH, *O and *OH), whose adsorption free energies (ΔG^{*OOH} , ΔG^{*O} and ΔG^{*OH}) have been identified as descriptors of ORR performance. The adsorption energies of multiple ORR intermediates are strongly related and difficult to be decoupled due to the scaling relations, thus the scaling relationships among the free energies of various reaction species (*OH, *O, and *OOH) can help simplify the search for highly active catalysts. Thereinto, as shown in supplementary Fig. 3., a good linear relationship can be observed between ΔG^{*OOH} and ΔG^{*OH} , namely, $\Delta G^{*OOH} = 1.27\Delta G^{*OH} + 2.63$. The slope of linear correlation between ΔG^{*OOH} and ΔG^{*OH} is close to unity since the electron transfer between both *OH, *OOH and Co center is nearly 1 electron. Nevertheless, we found that ΔG^{*OOH} tends to be less correlated with ΔG^{*OH} since their different bond order and the oxidation state of *O is not always -2 (please refer to *Angew. Chem. Int. Ed.* **2021**, 60, 16937-16941; *J. Phys. Chem. C* **2020**, 124, 13168-13176; *Appl. Surf. Sci.* **2022**, 592, 153237). Therefore, ΔG^{*OOH} can be represented by ΔG^{*OH} through the linear correlation and the overpotential can be determined by ΔG^{*OH} and ($\Delta G^{*O} - \Delta G^{*OH}$) values using the following equation:

$$\eta = \max \{ \Delta G^{*OOH} - 4.92, \Delta G^{*O} - \Delta G^{*OOH}, \Delta G^{*OH} - \Delta G^{*O}, -\Delta G^{*OH} \} / e + 1.23$$

It needs to be emphasized again that the entire ORR process involves four essential steps, each containing a proton-electron pair transfer. The step of oxygen adsorption (* + O₂ → *O₂) before the *OOH formation does not involve proton-electron transfer but is merely a chemical adsorption process. Thereby, the adsorption of O₂ molecules cannot be considered as the potential determining step, also the reaction energy barrier of this O₂ molecule adsorption step cannot be applied to prove the reaction thermodynamics. Instead, the reaction energy barrier of entire *OOH formation step (* + O₂ → *OOH) should be used for comparison. The free energy diagrams of oxygen adsorption step for all six models are downhill at U = 0 V, which indicates that the adsorption of O₂ molecules can proceed spontaneously, and it is thermodynamically favorable to adsorb O₂ prior to forming *OOH species.

2. When introducing functional groups, there's a significant deformation observed in the COP structure. Did the author assess the stability of their models?

Response: Thank you for your feedback. To obtain the accurate understanding of the

adsorption behaviors and foretell the ORR performance of cobalt porphyrin-based polymer electrocatalysts, we employed density functional theory (DFT) calculations to construct computational models.

Supplementary Fig. 1. The detailed model structures of (a) Por-H, (b) Por-CH₃, (c) Por-COCH₃, (d) Por-COOCH₃, (e) Por-COOH and (f) Por-CN.

The reviewer mentioned that when introducing functional groups, there's a significant deformation observed in the COP structure, this can be mainly attributed to the following aspects:

First, it is reported that many molecules can rotate internally around one or more of their bonds so that during a full 360° rotation, they will change between unstable and relatively stable conformations. Under the introduction of a rotation barrier, the staggered structure of polymers tends towards thermodynamic stability. Considering the rotation about the primary bond of the substituent, the substituents are generalized as spherical and the minimum van der Waals radius of substituent groups can normally be determined by the effective substituent width. Therefore, the discrepancy in the substituent groups and corresponding van der Waals radius will also lead to the differences in the configuration and size of polymers skeletons (please refer to *Nat. Chem.* **2012**, 4, 366–374; *Nature* **2001**, 411, 565–568; *J. Am. Chem. Soc.* **2016**, 138, 16639–16644; *Angew. Chem. Int. Ed.* **2021**, 60, 12742–12746).

Secondly, the influence of the steric nature of the substituent groups (–X, where X = H, CH₃, COCH₃, COOCH₃, COOH and CN) on the steric hindrance effect of the polymers should be considered. Thus, both the steric strain and steric hindrance can lead to minor deformation of polymers skeletons. In our DFT calculations, the thermodynamic stability of Por-X models with various substituents are ensured after the structural optimization. The thermodynamic stability of Por-X models after introducing large-steric-size substituents into the monomeric molecule has been widely verified from both experimental and theoretical point of view (please refer to *J. Am. Chem. Soc.* **2021**, 143, 30, 11423–11434; *J. Am. Chem. Soc.* **2020**, 142, 17524–17530; *Angew. Chem. Int. Ed.* **2021**, 60, 12742–12746).

In addition, the substituent steric interactions based on their size and shape will cause the ring strain and the corresponding bond angle geometry distortion, and the polymer skeletons in theoretical modeling tends to become negligibly twisted to

release the slight strain. In DFT calculation, these modeling structures are thermodynamically stable after the structure optimization converges, hence we are confident in the rationality and structural stability of our Por-X series models. This typical phenomenon can also be discovered in other published studies in the field of macrocycle-based materials (please refer to *Angew. Chem.* **2023**, 135, e202303871; *J. Am. Chem. Soc.* **2021**, 143, 2, 1098–1106; *Nat. Commun.* **2020**, 11, 5289)

3. In Fig. 1C, the authors designate the rate-determining step as (RDS). However, in the text, only the potential-determining step (PDS) is mentioned.

Response: Thank you for your constructive suggestion. We are very sorry for the negligence in our figures, and we have corrected this error in the revised manuscript.

Revised Fig. 1 | (c) The free energy diagram of Por-X analogs at $U = 0$ V.

4. Another crucial aspect is the author's determination of the rate-determining step (RDS). The desorption of $*OH \rightarrow * + OH^-$ is independent of the proton's activity. In contrast, the formation of $*OOH$, $*O$, $*OH$ depends on the proton concentration. These steps might be slower compared to the desorption of $*OH$, particularly when the proton activity is low in an alkaline environment. Consequently, determining the RDS solely on reaction energy could be misleading in this scenario.

Response: Thanks for your feedback. Free energy ladder diagram is one of the most intuitive methods to analyze the reaction mechanism, which enables dynamic analysis of the reaction process. The free energy diagram is able to assess the entire thermodynamic Gibbs free energy at the equilibrium potential and ultimately screen the reaction energy barriers for all elementary steps. The limiting reaction barrier is an important parameter that affects the catalytic properties, which can be evaluated by the free energy of PDS. It is well acknowledged that the PDS for ORR locates at the step with the smallest downhill energy (please refer to *Nat. Commun.* **2022**, 13, 57; *Nat. Commun.* **2022**, 13, 2075; *Nat. Commun.* **2021**, 12, 1734), also the $*OH$ desorption step is the PDS for many non-noble metal-based catalysts (please refer to *Angew. Chem. Int. Ed.* **2023**, 63, e202215441; *Nat. Commun.* **2018**, 9, 5422; *Adv. Energy Mater.* **2020**, 10, 2002896; *J. Am. Chem. Soc.* **2019**, 141, 20118–20126; *Angew. Chem. Int. Ed.* **2019**, 58, 18971–18980; *Angew. Chem. Int. Ed.* **2021**, 60, 27324–27329).

In our work, the theoretical calculation primarily proposes that the incorporation of electron-withdrawing substituents in the porphyrin secondary coordination sphere is

deemed to alleviate the overmuch binding strength of *OH species. As a result, the substituents with different electronic properties perform as regulator for varying the thermodynamic reaction barrier. Thereinto for Por-COOH model, the final *OH release step with the smallest downhill energy is regarded as the PDS, while the PDS of Por-CN model is of *OOH \rightarrow *O step since its excessive electron-withdrawing property changes the PDS, resulting in too weak or even unstable oxygen intermediate binding (Fig. 1c).

Revised Fig. 1 | (c) The free energy diagram of Por-X analogs at $U = 0$ V.

The reviewer claimed that the desorption of $*OH \rightarrow * + OH^-$ is independent of the proton's activity, while the formation of *OOH, *O, *OH depends on the proton concentration. However, as it is known, the dependence of ORR rates on the cations concentration/identity in alkaline media has been tracked, the saturation coverage of *OH is electrolyte-property dependent and it presumably interferes with the lateral interactions between *OH species (please refer to *Nat. Catal.* **2022**, 5, 615-623; *Adv. Sci.* **2020**, 7, 2000176). This can be rationalized by the fact that, in contrast to acidic solution, *OH specie is also more likely to adsorb at the active center due to the much higher OH^- concentration (much lower proton concentration) in alkaline electrolyte. Thereby, compared to the formation of *OOH, *O, *OH steps, the *OH release also plays a crucial role in determining the catalytic efficiency. Also, in our work, the secondary sphere microenvironment customization strategy reduced energy barrier of the potential-determining step (*OH desorption) in the reaction pathway, thus promoted the ORR efficiency.

5. Their Tafel slope is 41.1 (mV dec⁻¹). In a previous study (*ACS Catal.* 4, 4364–4376, 2014), it was indicated that a Tafel slope of 40 suggests 1~2 electrons transfer before the RDS. Consequently, the formation of *O could potentially be the RDS, contradicting their theoretical suggestion that the desorption of *OH is the RDS. Furthermore, their result show that there is a large change in the Tafel slope which suggest the change in the RDS. However, the calculation only shows the desorption of *OH as the RDS.

Response: Thank you for your suggestion and provide us the literature (*ACS Catal.* 4, 4364-4376, 2014). We have meticulously read this literature and found that our work is different from the mentioned literature. In this literature, the authors claimed that this Tafel kinetics study are limited to well-defined, nonreconstructed crystalline

planes together with pure water solvent, where the overall reaction system needs to be highly simple. Moreover, this literature mainly revolved around the investigation for the Tafel kinetics on noble metal surface. Besides, it is reported that the precise values of Tafel slope tend to vary a small amount with crystal facet and factors such as electrolyte purity, uncompensated resistance correction, and the chosen boundaries for linear fitting of the curve. Thereby, though this literature possesses great reference value for microkinetics, its conclusions do not suitable for our work.

According to many reported literatures, low Tafel slopes in the range of 30-50 mV dec⁻¹ have been reported for large surface area catalysts, as high surface area catalysts (if conductive and accessible) will already have a high geometric current density while the specific current density is still low and will be less limited (please refer to *Angew. Chem.* **2023**,135, e202216). Also from an experimental perspective, many studies presented similar or even lower Tafel values to demonstrate the intrinsic activity superiority of their catalysts, further verifying the rationality of Tafel slope of 40 mV dec⁻¹ in our work (please refer to *Nat. Commun.* **2020**, 11, 4173; *Angew. Chem. Int. Ed.* **2022**, 61, e202201104; *J. Am. Chem. Soc.* **2017**, 139, 17269–17272).

Thereby, relying solely on the Tafel value to judge the PDS is somewhat inaccurate, and we believe that our obtained Tafel values is scientific, which are derived through careful calculations.

Reviewer #3:

In this contribution, Huang et al. established the Sabatier plot by DFT calculation to forecast the catalytic O₂ reduction efficiency for a series of Co porphyrins with substitutions with different electron-withdrawing properties at the second coordination sphere of CoN₄ sites. It was shown that carboxyl group-substituted catalyst reduced the σ -bond orbital occupation, thus exhibiting an optimal OH* binding with Co center, which should be responsible for the best reactivity. The theoretical prediction was also well verified by rational design of different fringe groups modified Co porphyrins. Some related concerns are listed in the following:

Response: We sincerely appreciate your very positive reviews. The answers to the questions you raised are addressed in detail as below.

(1) I do not agree with viewpoint of PDS on the *OH desorption step for Co-N-C catalysts. Actually, the *OH desorption step is usually deemed as RDS step for Fe-N-Cs rather Co-N-Cs due to the strong binding of OH intermediates on Fe sites. In contrast, the relatively weak oxophilicity for Co metal leads to the difficulty in the O₂ activation and readily OOH* desorption. As such, the GOOH for Co-N-C is supposed to located at weak binding side of Volcanic plot for 4e- pathway. For 2e-pathway, the OOH* desorption should be RDS, thereby locating at the strong adsorption side of volcanic plot. Theoretically speaking, the Co-N-Cs should be governed by 2e- pathway, which seems to contradict with conclusions described in the work, please, give a rational explanation.

Response: Thanks for your feedback. As is known, for ORR the last hydrogenation step ($*\text{OH} \rightarrow \text{OH}^-$) appears to be the PDS for nearly all the atomically dispersed M-N-C catalysts due to the hybridization of metal d and O p orbitals. For M-N-C catalysts, the N atoms not only act as the anchoring sites to stabilize the single metal atoms but also play important roles in modulating the electronic structures of the active sites. Thereinto, the strong electronegativity of N atom can undesirably alter the electronic properties of metal catalytic center and increase the free adsorption energy of oxygen reaction intermediates (please refer to *J. Phys. Chem. C* **2020**, 124, 13168–13176; *Adv. Energy Mater.* **2020**, 10, 2002896; *J. Am. Chem. Soc.* **2019**, 141, 20118–20126). Among transition metal-based single-atom catalysts, only Ni atom is universally recognized as the active center with relatively weak oxygen binding affinity, while the other transition metals all exhibit different oxygen affinity in their corresponding electronic configuration microenvironment (please refer to *Nat. Commun.* **2023**, 14, 1792; *Adv. Funct. Mater.* **2023**, 33, 2210867).

The Co-N₄ configuration has been extensively explored due to its positioning at the top of volcano plot for the ORR limiting potential against the oxygen adsorption free energy. Further, through various optimization strategies, the intrinsic performance of Co-N₄ site has been enhanced. Among the literatures, many studies reported that the original Co-N₄ sites require modification to reduce the thermodynamic barrier for OH* desorption for further final product release, thus increasing the catalytic efficiency of catalysts (please refer to *J. Am. Chem. Soc.* **2019**, 141, 20118–20126; *J. Am. Chem. Soc.* **2017**, 139, 17269–17272; *Adv. Mater.* **2022**, 34, 2204021; *Adv. Mater.* **2019**, 31, 1901666; *Adv. Energy Mater.* **2020**, 10, 2002896; *Adv. Funct. Mater.* **2023**, 33, 2210867; *Chem. Eng. J.* **2022**, 430, 132642; *Chem. Eng. J.* **2022**, 429, 132119). Meanwhile, it is well-known that the binding of oxygen species at the porphyrin induces the charge transfer towards the oxygen intermediates, therefore the negatively charged and nucleophilic characters of the oxygen make it tightly bind to the positively charged active center.

In our work, the theoretical Sabatier volcano plot was primarily derived and forecasted the tough $*\text{OH}$ release on cobalt porphyrins and the incorporation of electron-withdrawing substituents on porphyrin secondary coordination sphere is deemed to alleviate the overmuch $*\text{OH}$ binding strength. On this basis, the secondary sphere microenvironment customization strategy was proposed to promote the catalytic efficiency of cobalt porphyrin-based polymers and experimentally validate the Sabatier principle. Thereinto, the carboxyl-substituted porphyrin model with an electron-deficient Co center is expected to approach the top of the Sabatier volcano map and emerges the highest ORR reactivity. From the experimental perspective, systematic characterizations have been applied to demonstrate the oxygen affinity of catalysts and track the dynamic evolution between metal center and oxygen intermediate for elucidating the catalytic mechanism.

As shown in revised **Fig. 6**, progressive testing techniques incorporating electrochemical measurements and *in situ* characterization approaches were firstly used to achieve a real-time analysis for ascertaining the intermediates absorbed at Co sites. The ATR-SEIRAS manifests the consumption of dissolved oxygen (**Fig. 6a, b**).

The *in situ* electrochemical Raman spectra under a series of applied bias voltages on CoCOP-COOH@KB (**Fig. 6c**) confirms that the *OH species acts as the key intermediate and the final *OH desorption step is the PDS for ORR on CoCOP-COOH@KB catalyst, which is in good agreement with DFT calculations.

Additionally, we supplemented the zero-field cooling temperature-dependent magnetic susceptibility measurements and electron paramagnetic resonance (EPR) spectra of CoCOP-COOH@KB, CoCOP-H@KB and CoCOP-CH₃@KB to verify the spin state of their Co centers. The average number of the unpaired Co 3d electrons acquired from the magnetic susceptibility plots (**Fig. 6d**) decreases along with the electron-withdrawing property of secondary sphere substituent increases. This result suggests that the strong electron-withdrawing/donating effect induces the Co 3d electronic spin state transition, and the Co 3d shells possess different electronic configurations, which are high spin state with $d_{xy}^2d_{xz}^2d_{yz}^1d_z^21d_{x^2-y^2}^1$ and medium spin state with $d_{xy}^2d_{xz}^2d_{yz}^2d_z^21$ (**Fig. 6e**). Further, the shape and intensity of EPR profile are impacted by the lone pair electrons onto the structural transition (**Fig. 6f**). The EPR spectrum shows an obvious signal at $g \approx 2.00$ for CoCOP-CH₃@KB and CoCOP-H@KB, which is attributed to a typical high spin Co²⁺ signal. As the enhancement of the electron-withdrawing property on substituent, the spin structure of catalytic Co sites changes from high spin (CoCOP-CH₃@KB and CoCOP-H@KB) to medium spin (CoCOP-COOH@KB). The EPR results consequently indicate the weakened O₂ adsorption affinity of CoCOP-COOH@KB, which is in agreement with the theoretical calculation. Thermodynamically, the relatively weak adsorption of *OH permits the rapid final *OH release, and the $d_{x^2-y^2}$ and d_{xy} orbitals are neglected due to their negligible interactions with oxygen intermediates. Compared with high spin Co centers within CoCOP-CH₃@KB, the decreased proportion of electrons in bonding orbital of medium spin Co²⁺ within CoCOP-COOH@KB leads to an alleviative Co/*OH interaction (**Fig.6g**), which is more favorable for propelling ORR cycles.

Thus, based on above results, it can be concluded that using both computational and experimental methods, the secondary sphere microenvironment customization strategy establishes the appropriate Co-oxygen binding to promote the catalytic efficiency of cobalt porphyrin-based polymers and experimentally validate the Sabatier principle. Thereinto, the incorporation of electron-withdrawing carboxyl substituents is deemed to alleviate the *OH binding strength and the electron-deficient Co center is expected to approach the top of the Sabatier volcano map and emerges the highest ORR reactivity.

The added discussion about Revised Fig. 6:

In the meantime, the electron spin configuration and unpaired electrons on Co ions within CoCOP-COOH@KB, CoCOP-H@KB and CoCOP-CH₃@KB counterparts were probed by the zero-field cooling temperature-dependent magnetic susceptibility and ex situ cryo (100 K) electron paramagnetic resonance (EPR) spectroscopy. The average number of the unpaired Co 3d electrons acquired from the magnetic

susceptibility plots (**Fig. 6d**) decreases along with the electron-withdrawing property of secondary sphere substituent increases. This result suggests that the strong electron-withdrawing/donating effect induces the Co 3d electronic spin state transition, and the Co 3d shells possesses different electronic configurations, which are high spin state with $d_{xy}2d_{xz}2d_{yz}1d_z^21d_x^2-y^21$ and medium spin state with $d_{xy}2d_{xz}2d_{yz}2d_z^21$ (**Fig. 6e**). Further, the shape and intensity of EPR profile are impacted by the lone pair electrons onto the structural transition (**Fig. 6f**). The EPR spectrum shows an obvious signal at $g \approx 2.00$ for CoCOP-CH₃@KB and CoCOP-H@KB, which is attributed to a typical high spin Co²⁺ signal. As the enhancement of the electron-withdrawing property on substituent, the spin structure of catalytic Co sites changes from high spin (CoCOP-CH₃@KB and CoCOP-H@KB) to medium spin (CoCOP-COOH@KB). The EPR results consequently indicates the weakened O₂ adsorption affinity of CoCOP-COOH@KB, which is in agreement with the theoretical calculation. Thermodynamically, the relatively weak adsorption of *OH permits the rapid final *OH release, and the $d_x^2-y^2$ and d_{xy} orbitals are neglected due to their negligible interactions with oxygen intermediates. Compared with high spin Co centers within CoCOP-CH₃@KB, the decreased proportion of electrons in bonding orbital of medium spin Co²⁺ within CoCOP-COOH@KB leads to an alleviative Co/*OH interaction (**Fig.6g**), which is more favorable for propelling ORR cycles.

Fig. 6 | The *in situ* electrochemical spectroscopic characterizations and schematic diagram of potential determining step transition. (a) Operando ATR-SEIRAS spectra for ORR at 500-2000 cm^{-1} , (b) potential dependent O-O stretching peak intensity, (c) potential-dependent *in situ* Raman spectra during ORR process over CoCOP-COOH@KB catalysts in 0.1 M KOH solution. (d) Magnetic susceptibility of CoCOP-COOH@KB, CoCOP-H@KB and CoCOP-CH₃@KB catalysts. M.S. represents medium spin, H.S. represents high spin. (e) 3d electron occupations in the M.S. and H.S. states of Co²⁺ in CoCOP-X@KB. (f) Low-temperature EPR spectra recorded at T = 100 K of CoCOP-COOH@KB, CoCOP-H@KB and CoCOP-CH₃@KB catalysts. (g) The orbital interactions between Co centers with different spin states and the potential determining *OH intermediates.

(2) The author should give the Co mass loading in the CoCOP-X@KB catalysts. Most importantly, considering that the 2e⁻ + 2e⁻ (O₂—H₂O₂; H₂O₂-H₂O) pathway is most likely occurring in the Co-based catalyst, the ORR activity test with different catalyst loading should be carried out to exclude the possible 2e⁻ reduction reaction. Actually, the 2e⁻ reduction pathway is readily covered when using enough high catalysts loading since generated H₂O₂ could be further reduced due to the long residue time inside thick catalytic layer. I noticed that the H₂O₂ yield is high up to 20% in some catalysts in the RRDE test, which further strengthens the possibility of 2e⁻ reduction.

Response: Thank you for your feedback. We also agree that Co mass loading data for all samples should be provided to ensure the catalyst loading, thereby the corresponding ICP data have been added and discussed in our revised manuscript (see Supplementary Table 1).

The added discussion: The Co contents, measured by inductively coupled plasma optical emission spectroscopy (ICP-OES), were similar in all the catalysts, which guaranteed that the effect of molecular level microenvironment customization on ORR reactivity of CoCOP-X@KB catalysts can be explored independently (Supplementary Table 1).

Supplementary Table 1. Mass contents of the metallic elements in CoCOP-X@KB.

Sample	Co contents (wt%)
CoCOP-COOH@KB	0.17
CoCOP-CH ₃ @KB	0.15
CoCOP-H@KB	0.14
CoCOP-COCH ₃ @KB	0.13
CoCOP-COOCH ₃ @KB	0.14
CoCOP-CN@KB	0.19

We also added the information of the exact catalyst loading in our revised supplementary information.

The added discussion: Next, 10 μL of the catalyst ink was deposited on a glassy carbon electrode and dried at room temperature, and the catalyst loading on glassy carbon electrode surface is $\sim 50 \mu\text{g}$ ($\sim 0.25 \text{ mg cm}^{-2}$) for RDE measurements.

We agree that the ORR activity test with different catalyst loadings should be carried out to exclude the $2e^-$ reduction reaction. Fig. R6 presents the disk current densities for CoCOP-COOH@KB with different catalyst loadings of about 25, 75, 200, 250, and $500 \mu\text{g}_{\text{catalyst}} \text{ cm}^{-2}$ on the RDE in O_2 -saturated 0.1 M KOH solution. All LSV curves show the same typical shape, but the ORR performance is observably enhanced along with the increase of catalyst loading. Under the low loading of 25 and $75 \mu\text{g}_{\text{catalyst}} \text{ cm}^{-2}$, CoCOP-COOH@KB exhibit well $E_{1/2}$ of about 0.81 V since its high specific surface area allows the catalyst film to be uniformly spread on the glassy carbon electrode. The low current density compared to higher loading is assigned to the very thin or even non-fully coated catalyst film on electrode, thus the adsorbed oxygen intermediate can rapidly outward spread and escape from ring electrode, which is not conducive to the later reduction steps. This normal phenomenon is consistent with previously reported catalysts that tend to favor the 4-electron pathway (please refer to *Appl. Catal. B Environ.*, **2017**, 206, 115–126; *Electrochem. Commun.*, **2011**, 13, 447–449). Further, the catalyst performance gradually enhances as the loading increases, and it can be found that under an appropriate loading range (more than $200 \mu\text{g}_{\text{catalyst}} \text{ cm}^{-2}$), the catalyst exhibits almost unchanged activity and diffusion limited current density. In order to avoid the catalyst film being too thick and the resultant difficult oxygen mass transport within catalyst during ORR (the catalyst films were normally reported to be too thick when the loading exceeds $800 \mu\text{g}_{\text{catalyst}} \text{ cm}^{-2}$), our ORR activity tests for all CoCOP-X@KB catalysts were all measured under the suggested loading range ($\sim 250 \mu\text{g}_{\text{catalyst}} \text{ cm}^{-2}$).

In addition, the reviewer said that the H_2O_2 yield is up to 20% in the RRDE test. Indeed, among M-N-C catalysts, Fe-based active center exhibits the highest TOF and selectivity for 4-electron ORR pathway. However, Fe-N-C catalysts are consistently less stable under long-term working conditions since Fe ion could produce highly oxidizing OH radicals which will attack active sites. To exclude the impact of Fe center and consequently increase the catalyst stability, other metal centers have been widely explored. Thereinto, the Co-N₄ configurations attract widespread interest due to its positioning at the top of volcano plot for the ORR limiting potential against the oxygen adsorption free energy. Although the Co-N-C catalysts are more stable, their catalytic performance were in general slightly inferior (please refer to *J. Phys. Chem. C* **2017**, 121, 17796–17817; *ACS Catal.* **2018**, 8, 5024–5031; *J. Am. Chem. Soc.* **2022**, 144, 14505–14516; *J. Am. Chem. Soc.* **2020**, 142, 17524–17530). In our work, the secondary sphere microenvironment customization strategy was proposed to elevate the performance of CoCOP-COOH@KB, including the catalytic activity and selectivity. Hence, considering the intrinsic inferior selectivity of Co-N-C materials, we merely claimed that the microenvironment customization strategy **ensures the significant selectivity promotion** of CoCOP-COOH@KB towards four-electron ORR pathway in the revised manuscript. Also, we should emphasize that our work

aims at the effect of the microenvironmental modifications on the adsorption behaviors of oxygen and intrinsic activity of Co-N₄ moieties for demystifying the relevant structure-property correlation, rather than only pursuing the highest activity and selectivity for ORR electroreduction.

Fig. R6. LSV curves obtained on a rotating glassy carbon electrode coated with CoCOP-COOH@KB catalysts at different mass loadings ($\mu\text{g cm}^{-2}$), recorded in O₂-saturated 0.1 M KOH solution.

(3) Is it Co¹⁺ or Co²⁺ as active sites for O₂ activation? Please specify the Co electron valence for the COCOP-X@KB, and most importantly, the analysis of CV curves is necessary for the clarity of catalytic active sites for this series of catalysts.

Response: Thank you for the constructive suggestions. First, in the determination of Co electron valence for the COCOP-X@KB nanocomposites, X-ray spectroscopic characterizations, such as X-ray photoelectron spectroscopy (XPS) and X-ray absorption near-edge structure (XANES) were applied.

Fig. 3a. The high-resolution Co 2p XPS spectra of the as-synthesized CoCOP-X@KB analogs.

Revised Fig. 3b. Normalized Co K-edge XANES spectra (up), and the spectra at the pre-edge region (bottom left), white-line rising edge region (bottom right) of CoCOP-COOH@KB, CoCOP-H@KB and CoCOP-CH₃@KB in comparison to a Co metal foil, CoO, and CoTPP serving as control samples.

For metallic macrocycle molecules, pristine Co-N₄ active sites are generally considered to approach +2 electron valence state, the local charge distribution and Co electron valence are correlated to the electronegativity of the corresponding substituents. In the high-resolution Co 2p spectrum (Fig. 3a), with the alternation of electron-withdrawing/donating properties on secondary coordination sphere electron, the peaks gradually move to higher binding energy varied with the increasing substituent electron-withdrawing property, indicating the lower electron density of Co for CoCOP-X@KB analogs. For further testifying the accurate valence state of optimal CoCOP-COOH@KB catalyst, the XANES spectra which are more sufficiently sensitive for Co element valence state is adopted. The Co K-edge XANES spectra (Fig. 3b) show that compared with CoCOP-H@KB, the enhanced absorption edge and white line intensity of CoCOP-COOH@KB indicate a higher than +2 oxidation state of the X-ray absorbing Co center, which originates from the electron transfer from the Co-N₄ site to the carboxyl substituent group.

It is known that the significant feature related to both O₂ activation and O₂ reduction is the generation of high-valent metal oxo intermediates in the catalytic cycle. Negatively charged deprotonated porphyrin have a strong σ -donating ability to stabilize high-valent metal ions, which are key intermediates involved in ORR (please refer to *Chem. Rev.* **2017**, 117, 3717-3797; *Acc. Chem. Res.* **2022**, 55, 6, 878-892; *Adv. Energy Mater.* **2021**, 11, 2100866; *ACS Catal.* **2016**, 6, 6429-6437). Thereby, Co²⁺ metal centers are normally regarded as the active sites for O₂ activation, and high-valent metal Co²⁺/Co³⁺ transition is generally thought to be involved in the O-O bond rupture and further reduction process. Based on this, CV curves in O₂-saturated 0.1 M KOH solution of the series catalysts are shown in Fig. R7 and the CV profiles

all exhibit sharp oxygen reductive peak. Meanwhile, the main feature of the CV recorded in N_2 -saturated solution is firstly ascribed to the appearance of the Co^{1+}/Co^{2+} transition redox peak (Ox.1, Red. 1) around 0.5 V. Also, at higher potentials, the obvious redox peak around 1.2 V (Ox.2, Red. 2) is associated with Co^{2+}/Co^{3+} transition. Therefore, the potential of Co^{1+} conversion is much lower than the onset potential for electrocatalytic ORR, which indirectly validates that Co^{2+} is the active site participating in the ORR. Also, many studies in the field of Co-N-C catalysts reported that the Co^{2+} ion is the resting state of the Co center, and once the electronegative O_2 bound to the Co atom, the Co^{3+} center will transiently form (*ACS Catal.* **2022**, 12, 8610–8622; *J. Am. Chem. Soc.* **2010**, 132, 8, 2655–2662), matching well with our results.

Fig. R7. CV curves in O_2 -saturated 0.1 M KOH solution of (a) CoCOP- CH_3 @KB, (b) CoCOP-H@KB, (c) CoCOP-CO CH_3 @KB, (d) CoCOP-COO CH_3 @KB, (e) CoCOP-COOH@KB and (f) CoCOP-CN@KB.

Fig. R8. CV curves in N_2 -saturated 0.1 M KOH solution of CoCOP-COOH@KB at the voltage range from 0.3-0.9 V and 0.9-1.6 V.

(4) The limited diffusion current shown in the ORR measurement is low (ca. 4 mA/cm^2), please try to give a reasonable explanation.

Response: Thanks for your feedback. On the basis of mechanism, the intrinsic reactivity and conductivity of catalytic materials are determined as the decisive

factors for the value of limited diffusion current towards ORR. Thereinto, the rate of electrocatalytic ORR is greatly dependent on the electronic conductivity of the materials. The enhancement of conductivity can promote the electron transfer from the electrode to oxygen-containing species and increase the energy conversion efficiency towards ORR, thereby leading to an increase in limiting current value. The highly conductive conjugated skeleton enables fast electron transport throughout the backbones, and the favorable electrical property is normally attributed to high extent of graphitization of catalysts. This is the reason why synthesis routes for the majority of single-atom catalysts necessitate pyrolysis treatment to enhance their graphitization degree for better activity. Porphyrins molecule are tetrapyrrolic 18- π electron macrocycles with a largely conjugated aromatic ring, in our work we additionally enhance the electronic mobility within the overall backbones by constructing porphyrin-based conjugated polymers. However, from a structural perspective, when comparing to pyridine-type nitrogen-doped carbon materials with excellent electrical property, the inferior in-plane extended π -conjugation and higher pyrrolic N content of porphyrin somehow affects its electrical conductivity. This typical phenomenon can also be discovered in other published studies in the field of porphyrin-based polymer materials (please refer to *Angew. Chem. Int. Ed.* **2021**, 60, 8472-8476; *J. Am. Chem. Soc.* **2014**, 136, 6348–6354; *Adv. Energy Mater.* **2021**, 11, 2102062; *Angew. Chem. Int. Ed.* **2018**, 57,12567 –12572).

The reason for electing porphyrin as the building block is that their physical and chemical properties can be fine-tuned by suitable structural modifications to fulfill the requirements of low overpotential for different functional device. Therefore, porphyrin-based polymer is considered as a customized platform to explore enhancement origin of the ORR activity and establish the structure-performance relation for efficient M-N-C catalysts. Resultingly, in our work, the secondary sphere microenvironment customization strategy was proposed and the rapid interfacial/outward charge migration kinetics for the optimal CoCOP-COOH@KB catalyst is regarded as one of the most important factors contributing to its superior intrinsic reactivity.

In conclusion, this work aims at the effect of the microenvironmental modifications on the adsorption behaviors of oxygen and intrinsic activity of Co-N₄ moieties. We believe that our comprehensive experimental and theoretical results can draw this conclusion. In future, we also believe some strategies, such as molecular design, loading highly conductive supports can resolve the insufficient intrinsic conductivity of these molecules to further enhance the diffusion performance of pyrolysis-free materials.

REVIEWER COMMENTS

Reviewer #1 (Remarks to the Author):

This research controlled the oxygen reduction activity of single atomic Co active sites using electron-withdrawing/donating functionality at the second shell. Additionally, a unique method was employed to measure the catalyst's site density and confirm the high intrinsic activity of Co single-atomic catalysts. The originality of this paper lies in systematically confirming the performance changes of the catalyst due to electron-withdrawing/donating effects and demonstrating reaction mechanisms and catalyst characteristics through various analyses. However, additional analysis seems necessary to explain the performance improvement. Therefore, this paper requires minor revision for publication in Nature Communications.

Reviewer #2 (Remarks to the Author):

In the manuscript., the authors reported a theoretical descriptor based on the binding energies of oxygen adsorbates, which is associated with the derived Sabatier volcano plot. The authors introduced several functional groups aimed at altering the binding energies of adsorbates. These functional groups work by withdrawing electrons from the metal center, thereby helping to lower the σ -bond. The authors validated this using both computational methods and XANES spectra.

The modulation of the microenvironment is a new area of study as scientists try to mimic the natural enzymes, where functional groups could significantly contribute to altering the local environment of the active site. For instance, they can influence local pH, the diffusion of reactants and products, proton transfer, and more. This manuscript solely concentrates on the impact of functional groups on the valence electrons of the metal center, which has been reported in several papers, such as ACS Catal. 12, 7278-7287, 2022. Utilizing binding energies as a descriptor is not a novel concept either (Nature Catalysis, 5, 615–623, 2022; J. Phys. Chem. Lett. 2018, 9, 588–595; Chem. Rev. 2018, 118, 2302–2312 ; Adv.Sci. 7, 1901614, 2020).

Some technical questions:

1. Why did the author omit the first step (adsorption of $*O_2$) in the calculation of the overpotential, as depicted in Fig. S3?
2. When introducing functional groups, there's a significant deformation observed in the COP structure. Did the author assess the stability of their models?

3. In Fig. 1C, the authors designate the rate-determining step as (RDS). However, in the text, only the potential-determining step (PDS) is mentioned.

4. Another crucial aspect is the author's determination of the rate-determining step (RDS). The desorption of $*OH \rightarrow * + OH^-$ is independent of the proton's activity. In contrast, the formation of $*OOH$, $*O$, $*OH$ depends on the proton concentration. These steps might be slower compared to the desorption of $*OH$, particularly when the proton activity is low in an alkaline environment. Consequently, determining the RDS solely on reaction energy could be misleading in this scenario.

5. Their Tafel slope is 41.1 (mV dec⁻¹). In a previous study (ACS Catal. 4, 4364–4376, 2014), it was indicated that a Tafel slope of 40 suggests 1~2 electrons transfer before the RDS. Consequently, the formation of $*O$ could potentially be the RDS, contradicting their theoretical suggestion that the desorption of $*OH$ is the RDS. Furthermore, their result show that there is a large change in the Tafel slope which suggest the change in the RDS. However, the calculation only shows the desorption of $*OH$ as the RDS.

Reviewer #3 (Remarks to the Author):

In this contribution, Huang et al. established the Sabatier plot by DFT calculation to forecast the catalytic O₂ reduction efficiency for a series of Co porphyrins with substitutions with different electron-withdrawing properties at the second coordination sphere of CoN₄ sites. It was shown that carboxyl group-substituted catalyst reduced the σ -bond orbital occupation, thus exhibiting an optimal OH* binding with Co center, which should be responsible for the best reactivity. The theoretical prediction was also well verified by rational design of different fringe groups modified Co porphyrins. Some related concerns are listed in the following:

(1) I do not agree with viewpoint of PDS on the $*OH$ desorption step for Co-N-C catalysts. Actually, the $*OH$ desorption step is usually deemed as RDS step for Fe-N-Cs rather Co-N-Cs due to the strong binding of OH intermediates on Fe sites. In contrast, the relatively weak oxophilicity for Co metal leads to the difficulty in the O₂ activation and readily OOH* desorption. As such, the GOOH for Co-N-C is supposed to located at weak binding side of Volcanic plot for 4e⁻ pathway. For 2e⁻ pathway, the OOH* desorption should be RDS, thereby locating at the strong adsorption side of volcanic plot. Theoretically speaking, the Co-N-Cs should be governed by 2e⁻ pathway, which seems to contradict with conclusions described in the work, please, give a rational explanation

(2) The author should give the Co mass loading in the CoCOP-X@KB catalysts. Most importantly, considering that the 2e⁻ + 2e⁻ (O₂—H₂O₂; H₂O₂-H₂O) pathway is most likely occurring in the Co-based catalyst, the ORR activity test with different catalyst loading should be carried out to exclude the possible 2e⁻ reduction reaction. Actually, the 2e⁻ reduction pathway is readily covered when using enough high catalysts loading since generated H₂O₂ could be further reduced due to the long

residue time inside thick catalytic layer. I noticed that the H₂O₂ yield is high up to 20% in some catalysts in the RRDE test, which further strengthens the possibility of 2e⁻ reduction.

(3) Is it Co¹⁺ or Co²⁺ as active sites for O₂ activation? Please specify the Co electron valence for the COCOP-X@KB, and most importantly, the analysis of CV curves is necessary for the clarity of catalytic active sites for this series of catalysts.

(4) The limited diffusion current shown in the ORR measurement is low (ca. 4 mA/cm²), please try to give a reasonable explanation.

Manuscript NCOMMS-23-53238-point-by-point responses to the reviewer's comments

We thank the reviewers for the constructive comments which have helped us to greatly improve our research and the quality of our manuscript. We have now included additional analysis and discussions to fully address the reviewers' concerns and suggestions. Furthermore, we have modified our Manuscript and Supporting Information based on the additional results and analysis, which are highlighted in red. Below, we address the points raised by reviewers one by one.

Reviewer #1: This research controlled the oxygen reduction activity of single atomic Co active sites using electron-withdrawing/donating functionality at the second shell. Additionally, a unique method was employed to measure the catalyst's site density and confirm the high intrinsic activity of Co single-atomic catalysts. The originality of this paper lies in systematically confirming the performance changes of the catalyst due to electron-withdrawing/donating effects and demonstrating reaction mechanisms and catalyst characteristics through various analyses. However, additional analysis seems necessary to explain the performance improvement. Therefore, this paper requires minor revision for publication in *Nature Communications*.

Response: We appreciate your very positive review and recommendation that our manuscript could be accepted for publication in *Nature Communications* after minor revision. The answers to the questions you raised are addressed in detail as below.

1. According to the author's DFT calculations, it was claimed that Fe and Co sites exhibit a similar trend with OH* strong adsorption energy, while Ni sites show the OH* weak adsorption energy. To accurately support the experimental results that Co sites has strong binding with oxygen species in the ORR process, it is necessary to demonstrate whether introducing electron-withdrawing/donating groups into FeCoP-based or NiCoP-based catalysts.

Response: We would like to thank the reviewer for the constructive comment, and we agree with the reviewer's point. We extended the microenvironment customization strategy to other representative metal centers and proved the universality on optimizing O₂ adsorption affinity of metalloporphyrin catalysts. By simply replacing cobalt source with iron source and nickel source, Fe porphyrin-based (FeCOP-CH₃@KB, FeCOP-H@KB, FeCOP-COOH@KB) and Ni porphyrin-based (NiCOP-CH₃@KB, NiCOP-H@KB, NiCOP-COOH@KB) polymer nanocomposites were prepared using identical synthetic procedures.

Typical scanning electron microscopy (SEM) and the corresponding energy dispersive X-ray spectroscopy (EDS) mapping images of Fe porphyrin-based and Ni porphyrin-based polymer nanocomposites, as shown in Fig. R1-R6, prove the similar blocky micron-sized particle structures as CoCOP-X@KB materials. The EDS mapping images show the homogenous distribution of C, N and corresponding O, Fe,

Ni elements in the as-prepared Fe and Ni-based nanocomposites, representing the universality on synthesis route of secondary sphere microenvironment customization strategy.

Fig. R1. The SEM image and corresponding elemental mapping images of FeCOP-CH₃@KB.

Fig. R2. The SEM image and corresponding elemental mapping images of FeCOP-H@KB.

Fig. R3. The SEM image and corresponding elemental mapping images of FeCOP-COOH@KB.

Fig. R4. The SEM image and corresponding elemental mapping images of NiCOP-CH₃@KB.

Fig. R5. The SEM image and corresponding elemental mapping images of NiCOP-H@KB.

Fig. R6. The SEM image and corresponding elemental mapping images of NiCOP-COOH@KB.

LSV curves at various rotational speeds (from 400 to 1600 rpm) of these six catalysts are shown in Fig. R7, R8, and the essential catalytic parameters are summarized in Table R1. According to Sabatier principle, the interaction between the catalytic sites and the adsorbates should be neither too strong nor too weak, too weak interaction

brings low reduction reaction rate, meanwhile too strong interaction may cause surface oxidization and hampers the subsequent transform process or even desorption. Hereinto Fe-based and Co-based centers strongly interact with oxygen and facilitate the O₂ adsorption, however the release of OH* requires overcoming the high reaction barrier, which means the introduction of electron-withdrawing substituents on porphyrin second sphere will improve the catalytic performance. On the contrary, the first electron transfer (O₂ → OOH*) is generally the PDS on Ni centers, so the modification of electron-donating substituents may enhance the intrinsic activity (please refer to *Adv. Mater.* **2017**, 29, 1606635; *Adv. Mater.* **2021**, 34, 2104891).

As expected, compared to FeCOP-H@KB electrode, a positive shift ($\Delta E = 60$ mV) of half-wave potential on FeCOP-COOH@KB electrode can be observed, demonstrating the boosted electrocatalytic activity of electron withdrawing substituent-modified FeCOP-COOH@KB towards ORR induced by microenvironment customization strategy. Similarly, electron donating substituent-modified NiCOP-CH₃@KB shows improved performance compared to NiCOP-H@KB. For NiCOP-H@KB and NiCOP-COOH@KB, the very wide voltage range of the mixed kinetic- and diffusion-controlled region is attributed to their electro-kinetic retardation, implying the unsatisfactory intrinsic activity of original Ni sites with weak oxygen affinity.

Therefore, the microenvironment customization strategy developed in our study is universal to prepare M-N-C materials with different transition metals as center atoms, and customize the adsorption behaviors of oxygen intermediates for accelerating the oxygen reduction electrocatalysis.

Fig. R7. RDE LSV curves at different rotation speeds of (a) FeCOP-CH₃@KB, (b) FeCOP-H@KB and (c) FeCOP-COOH@KB.

Fig. R8. RDE LSV curves at different rotation speeds of (a) NiCOP-CH₃@KB, (b) NiCOP-H@KB and (c) NiCOP-COOH@KB.

Table R1. The summary of E_{onset} (V vs. RHE), $E_{1/2}$ (V vs. RHE) and diffusion limited current density (J_L , mA cm⁻²) values of FeCOP-CH₃@KB, FeCOP-H@KB, FeCOP-COOH@KB, NiCOP-CH₃@KB, NiCOP-H@KB and NiCOP-COOH@KB catalysts.

Sample	E_{onset}	$E_{1/2}$	J_L
FeCOP-CH ₃ @KB	0.85	0.74	5.1
FeCOP-H@KB	0.88	0.74	5.0
FeCOP-COOH@KB	0.90	0.80	4.5
NiCOP-CH ₃ @KB	0.83	0.72	5.1
NiCOP-H@KB	0.88	0.70	5.1
NiCOP-COOH@KB	0.89	0.68	4.8

2. In figure 3b, the authors presented the change in oxidation states using XANES. However, it is anticipated that if XANES peaks are observed in energy ranges lower than the presented interval, the data will overlap and the trends will not align with the performance data. Additionally, it is necessary to add a reference for Co³⁺. It would be better to show the derivated XANES spectra. Authors showed fitting data. It is considered that the fine fitting data when R-factor was lower than 0.02. Please show more accurate fitting data.

Response: Thanks for your constructive suggestion. We do admit that the difference in the observed energy range of Co K-edge XANES spectra will impact the judgment on the oxidation states change, which exists widely in many previously reported studies (please refer to *Nat. Commun.* **2022**, 13, 723; *Adv. Mater.* **2022**, 34, 2204021; *J. Am. Chem. Soc.* **2019**, 141, 20118; *Angew. Chem. Int. Ed.* **2021**, 60, 16937; *Nat. Commun.* **2023**, 14, 172). Therefore, the oxidation states of CoCOP-CH₃@KB, CoCOP-H@KB and CoCOP-COOH@KB catalysts are only compared in the rising region of XANES spectra. Thereinto, the peak position of CoCOP-COOH@KB shifts toward higher energy due to the 2p block stabilization, CoCOP-COOH@KB possesses the most positive charged Co center, which matches well with the result in XPS.

Also, we agree that the first derivative of Co K-edge XANES spectra is very important for verifying the effect of secondary sphere substituents on molecular level microenvironment customization. Determining the edge position is an efficient approach to evaluate the oxidation state of Co elements, which can be achieved by identifying the intensity maximum of the first derivative of XANES spectra. Thus, the first derivative XANES spectra and average oxidation state determination by linear fitting calibration from XANES spectra were added in our revised manuscript (Revised **Fig. 3b**). Moreover, we replenished the Co₂O₃ as the reference for Co³⁺ oxidation state and the corresponding discussion have been added in our revised manuscript.

Thanks for your professional and helpful suggestions on the EXAFS fitting data. We reanalyzed the quantitative structural parameters around Co atoms and reperformed the curve fitting for further improving the accuracy of our provided EXAFS fitting data. The EXAFS fitting curves (Fig. R9) and quantitative fitting of the EXAFS spectra (Table R2) show the first coordination shell of four Co-N bonds at the similar distance

of about 1.9 Å, and the second sphere by Co-C bonds is clearly demonstrated at the distance of 2.9 Å. Specifically, error bounds (goodness of fit) that describe the structural parameters obtained by EXAFS spectra have been optimized to achieve the R-factor lower than 0.02. Together, CoCOP-CH₃@KB, CoCOP-H@KB and CoCOP-COOH@KB all adopt identical coordination configuration as Co-N₄ moiety.

The added discussion about Revised Fig.3b:

Compared with CoCOP-CH₃@KB and CoCOP-H@KB counterparts, the Co K-edge XANES spectra edge energy (~7,730 eV) for CoCOP-COOH@KB shows higher peak intensity, which represents a higher oxidation state of Co center in CoCOP-COOH@KB, consistent with the XPS results. To display the accurate oxidation states of Co atoms, the first derivative curve and average valence states determination by linear fitting calibration of the Co K-edge XANES were carried out. The first derivative curve exhibits that the E_0 value (calculated as the highest inflection point on the absorption edge) of CoCOP-COOH@KB (7723.5 eV) is situated between CoO (7721.1 eV) and Co₂O₃ (7725.7 eV). The linear fitting curves were obtained for CoCOP-COOH@KB, CoCOP-H@KB, CoCOP-CH₃@KB catalysts, and Co foil, CoO, Co₂O₃ serving as control samples. The linear fitting result suggests the different Co oxidation states of CoCOP-COOH@KB, CoCOP-H@KB and CoCOP-CH₃@KB ranging from +2 (CoO) to +3 (Co₂O₃). The average Co oxidation state of CoCOP-COOH@KB (+2.6) is the highest among three catalysts, which originates from the electron transfer from the Co-N₄ site to the carboxyl group.

Fig. 3 | Electronic states and atomic structure analysis of Co atoms in CoCOP-X@KB analogs. (a) The high-resolution Co 2p XPS spectra of the as-synthesized CoCOP-X@KB analogs. (b) Normalized Co K-edge XANES spectra (up), first derivative of Co K-edge XANES spectra (bottom left), and average oxidation state

determination by linear fitting calibration of CoCOP-COOH@KB, CoCOP-H@KB and CoCOP-CH₃@KB in comparison to Co foil, CoO, and Co₂O₃ control samples. (c) The Fourier transformed magnitude of the k³-weighted Co K-edge EXAFS spectra of CoCOP-COOH@KB, CoCOP-H@KB, CoCOP-CH₃@KB, Co foil, CoO, and CoTPP. (d) WT-EXAFS plots of CoCOP-COOH@KB, Co foil and CoTPP.

Fig. R9. EXAFS fitting curves in the R -space of (a) CoCOP-CH₃@KB, (b) CoCOP-H@KB and (c) CoCOP-COOH@KB.

Table R2. EXAFS fitting parameters at the Co K-Edge for CoCOP-COOH@KB, CoCOP-H@KB and CoCOP-CH₃@KB.

Sample	Shell	CN	R(Å)	$\Delta\sigma^2 \cdot 10^3$ (Å ²)	R-factor
CoCOP-COOH@KB	Co-N	4.0	1.91	1.2	0.019
	Co-C	7.1	2.91	12.0	
CoCOP-H@KB	Co-N	3.9	1.91	1.5	0.012
	Co-C	7.7	2.89	13.9	
CoCOP-CH ₃ @KB	Co-N	4.1	1.93	10.4	0.016
	Co-C	7.3	2.92	3.4	

3. In Figure 6a-c, in-situ Raman data according to potential was shown, showing that OH* species acts as a key intermediate in the reaction. I recommend comparing in-situ Raman data with CoCoP-H@KB and CoCoP-CH₃@KB catalysts to check OH* desorption changes to compare mechanisms.

Response: Thank you for this thought-provoking suggestion. To monitor the evolution of catalyst structure and the surface adsorbates behaviors of CoCOP-H@KB and CoCOP-CH₃@KB counterparts during ORR, *in situ* Raman spectra were conducted and visualized with contour plots (Supplementary Fig. 59), also the corresponding discussion have been added in our revised manuscript. As shown in Fig. 6c, the characteristic peak intensity of the oxygen species on PDS gradually decreases when the potential decreases on the CoCOP-COOH@KB electrode. Differently, the symmetric stretching signals of Co-OOH, Co-O, Co-OH observed on CoCoP-H@KB and CoCoP-CH₃@KB electrodes generally increase along with the potential decreases, representing the blockage of Co centers by strongly adsorbed oxygen species (please refer to *Angew. Chem. Int. Ed.* **2019**, 58, 10677-10682; *Adv. Funct. Mater.* **2023**, 33,

2210867). We believe that these results can convincingly prove the high affinity toward oxygen species of original porphyrinic Co-N₄ sites during ORR, and the improved ORR kinetics of CoCOP-COOH@KB can be partly attributed to the rapid OH⁻ release from Co-N₄ sites.

The added discussion about Revised Supplementary Fig. 59:

For CoCOP-H@KB electrode (Supplementary Fig. 59a), the peak at around 600 and 690 cm⁻¹ were assigned to the symmetric stretching of $\nu(\text{Co-OOH})$ and $\nu(\text{Co-O})$, respectively. As the potential decreases, $\nu(\text{Co-OOH})$ signal weakens when the $\nu(\text{Co-O})$ increases as the potential reaches 0.5 V, which is originated from the blockage of Co centers by subsequent firmly adsorbed *OH species. Moreover, the peaks at 780 cm⁻¹ for CoCOP-CH₃@KB electrode (Supplementary Fig. 59b), which represents the $\nu(\text{Co-OH})$ signal, further confirms the overmuch oxygen binding strength on electron-donating group modified Co-N₄ center. Based on the results presented, we conclude that the porphyrin-based Co-N₄ centers normally possess high affinity toward O₂ species, the *OH species acts as the key intermediate and the final *OH desorption step is the PDS for ORR on original porphyrinic Co-N₄ active sites. Thereinto, the carboxyl substituent accelerates the ORR kinetics on CoCOP-COOH@KB by relieving the overmuch binding strength, which is in good agreement with DFT calculations. These findings afford a clear perception of the elementary processes over CoCOP-COOH@KB catalysts, and the corresponding ORR mechanism is proposed in Supplementary Fig. 60.

Supplementary Fig. 59. Contour-type *in situ* Raman spectra of (a) CoCOP-H@KB electrode and (b) CoCOP-CH₃@KB electrode.

Reviewer #2: While we agree that the adsorption of O₂ may not be the RDS, the lower energy of *O₂ will increase the reaction barrier for the subsequent step (as opposed to comparing it with the gas phase energy of 4.92 eV). Therefore, if the adsorption of O₂ is considered in the reaction pathway (O₂ (g) → *O₂ → OOH* → O* → OH* → OH⁻), it must be included in the calculation of the overpotential. Consequently, the

overpotential should be calculated as follows:

$$\eta = \max \{ \Delta G^*_{\text{OOH}} - \Delta G^*_{\text{O}_2}, \Delta G^*_{\text{O}} - \Delta G^*_{\text{OOH}}, \Delta G^*_{\text{OH}} - \Delta G^*_{\text{O}}, -\Delta G^*_{\text{OH}} \} / e + 1.23$$

$\Delta G^*_{\text{O}_2}$ is the Gibbs free energy of adsorbed $^*\text{O}_2$ (i.e., 4.31 eV in Fig S4).

The author's method to calculate overpotential only makes sense when the reaction steps are considered as $\text{O}_2(\text{g}) \rightarrow \text{OOH}^* \rightarrow \text{O}^* \rightarrow \text{OH}^* \rightarrow \text{OH}^-$ as mentioned by author's references (*Nat Commun* 9, 5422 (2018); *Angew. Chem. Int. Ed.* 2021, 60, 16937-16941). In that case, the adsorption of O_2 was not considered in the reaction pathway. Additionally, there is no point in including the $^*\text{O}_2$ adsorption step if it is subsequently ignored in the author's calculation of overpotential.

Finally, the author should specify the reaction equation for each step (i.e., as seen in *Nat Commun* 9, 5422 (2018)), at least in the supporting information.

Response: Thanks for your very constructive suggestion. As we have elaborated, only the Gibbs free energy of elementary steps involving proton-electron transfer process can be employed to gain the theoretical overpotential. Namely, as the reviewer mentioned, the calculated overpotential only makes sense when the reaction steps are considered as $\text{O}_2(\text{g}) \rightarrow \text{OOH}^* \rightarrow \text{O}^* \rightarrow \text{OH}^* \rightarrow \text{OH}^-$. In the previous version of our manuscript, we introduced the Gibbs free energy of adsorbed $^*\text{O}_2$ to access the ease of oxygen reactants' adsorption on catalytic Co-N₄ sites, and analyzed whether it can spontaneously chemisorb on the active center to undergo subsequent proton-electron transfer processes.

Considering the valuable feedback from the reviewer, we concerned that the value of oxygen adsorption energy of $^*\text{O}_2$ may lead to some confusions or misunderstandings in reading. Therefore, we have redrawn the corresponding free energy diagrams (Fig. 1c and Supplementary Fig. 4-9) to explore the ORR processes on Co-N₄ configurations more clearly from a thermodynamic perspective in the revised manuscript.

We also agree that the ORR reaction equation for each step should be provided, thereby the overall reaction equation and corresponding elementary step equations have been added in our revised manuscript.

The added discussion:

The overall reaction of ORR occurring in the alkaline conditions is: $\text{O}_2 + 2\text{H}_2\text{O} + 4e^- = 4\text{OH}^-$. The ORR proceeds through the following elementary steps, which are generally used to prove the ORR electrocatalytic process on active sites:

where * represents the active site on catalytic surface, (l) and (g) stand for liquid and

gas phases, respectively.

Revised Fig. 1 | (c) The free energy diagram of Por-X analogs at $U = 0$ V.

Revised Supplementary Fig. 4. The free energy diagram with four essential steps of Por-CH₃ model at $U = 0$ V.

Revised Supplementary Fig. 5. The free energy diagram with four essential steps of Por-H model at $U = 0$ V.

Revised Supplementary Fig. 6. The free energy diagram with four essential steps of Por-COCH₃ model at U = 0 V.

Revised Supplementary Fig. 7. The free energy diagram with four essential steps of Por-COOCH₃ model at U = 0 V.

Revised Supplementary Fig. 8. The free energy diagram with four essential steps of Por-COOH model at U = 0 V.

Revised Supplementary Fig. 9. The free energy diagram with four essential steps of Por-CN model at $U = 0$ V.

Reviewer #3: I have read in details the author's point-to-point responses to my questions, and I feel now that the manuscript could be considered as publication in *Nature communications* with a minor concern: In Fig. R6, the author should provide the corresponding data for ring current and give the H₂O₂ selectivity as well as transferred electric number, especially in the low catalysts loading (i.e. 25 mg/cm² and 75 mg/cm²).

Response: We appreciate your very positive review and the recommendation that our manuscript could be accepted for publication in *Nature Communications* after minor revision.

We agree that the RRDE results including the disk current and ring current are very important to exclude the possible 2-electron reduction reaction over catalysts. Thereby, the ring currents and disk currents of CoCOP-COOH@KB electrode based on the RRDE voltammograms were obtained. Fig. R10 shows that the RRDE results for CoCOP-COOH@KB with different catalyst loadings of about 25, 75, 250, and 500 $\mu\text{g}_{\text{catalyst}} \text{cm}^{-2}$ all exhibit similar curve shape. Still, at a fixed electrode rotation rate, the onset potential, half-wave potential and diffusion limited current density are all monotonically enhanced along with the catalyst mass loading increases within the range lower than 250 $\mu\text{g}_{\text{catalyst}} \text{cm}^{-2}$. The amount of H₂O₂ (%H₂O₂) released by the oxygen reduction on the disk into the electrolyte was monitored at the platinum ring, significantly, the %H₂O₂ generated heavily depends on the amount of the catalyst deposited on the disk electrode. For CoCOP-COOH@KB, the %H₂O₂ gradually decreases along with the catalyst mass loading on the disk. When the catalyst mass loading is very low (such as 25 $\mu\text{g}_{\text{catalyst}} \text{cm}^{-2}$ and 75 $\mu\text{g}_{\text{catalyst}} \text{cm}^{-2}$), the relatively higher %H₂O₂ and lower diffusion limited current density can be observed. This can be attributed to the very thin catalyst film on the electrode surface, so that the incomplete 2 electron product H₂O₂ prefers to suddenly outward transfer and escape from ring electrode into bulk electrolyte, rather than undergoing the further reduction to OH⁻. Additionally, due to the irregular non-fully coated distribution of the catalyst film, the turbulences in electrolyte flow near the overall electrode surface are inevitable,

resulting in a decreased retention time of H₂O₂ on the disk electrode (please refer to *Catal. Today*, **2016**, 262, 74–81; *Appl. Catal. B Environ.*, **2017**, 206, 115–126; *Electrochem. Commun.*, **2011**, 13, 447–449).

To ensure the catalysts film can be enough to cover the whole electrode surface, the mass loading is gradually increased. Resultingly, the H₂O₂ generated inside the catalyst film diffuses throughout the electrode and further react on other active sites, which is prior to its release into bulk electrolyte and subsequent detection by Pt outer ring (please refer to *Electrochem. Solid-State Lett.*, **2008**, 11, B105; *Electrochem. Commun.*, **2008**, 10, 611–615). Therefore, under the suggested loading (~250 μg_{catalyst} cm⁻²), CoCOP-COOH@KB exhibits the average %H₂O₂ less than 6% and the transferred electron number greater than 3.5, demonstrating that it favors the direct 4 electron reaction pathway. The obtained values of these catalytic parameters for CoCOP-COOH@KB under different catalyst loadings are listed in Table R3. Hence, the selectivity promotion of CoCOP-COOH@KB towards 4 electron ORR pathway by microenvironment customization strategy was confirmed by the values of %H₂O₂ and transferred electron number, especially average transferred electron number > 3.2, regardless of the catalyst mass loading.

Fig. R10. (a) RRDE voltammograms of CoCOP-COOH@KB catalysts at different mass loadings (μg cm⁻²) recorded in oxygen-saturated 0.1 M KOH at the rotation speed of 1600 rpm. (b) The electron transfer number and the amount of H₂O₂ produced during ORR on CoCOP-COOH@KB electrode.

Table R3. The obtained values of E_{onset} (V vs. RHE), $E_{1/2}$ (V vs. RHE), diffusion limited current density (J_L , mA cm⁻²), ORR electron transferred number, and %H₂O₂ produced during ORR for CoCOP-COOH@KB under different catalyst loadings (μg_{catalyst} cm⁻²).

Mass loading	E_{onset}	$E_{1/2}$	J_L	transferred number	%H ₂ O ₂
25	0.81	0.77	2.5	3.0-3.4	10.3-18.6
75	0.87	0.82	3.2	3.2-3.4	8.9-13.9
250	0.91	0.86	4.0	3.5-3.6	6.2-8.9
500	0.92	0.86	4.2	3.5-3.6	6.0-8.9

REVIEWERS' COMMENTS

Reviewer #1 (Remarks to the Author):

Authors well explained about the reviewer's comments and make up the additional analysis data for mechanistic study. The originality of this paper lies in systematically confirming the performance changes of the catalyst due to electron-withdrawing/donating effects and demonstrating reaction mechanisms and catalyst characteristics through various analyses. Although, we have minor concern, this paper could be considered for publication in Nature Communications.

1. The authors demonstrated a high site density and turnover frequency of the synthesized catalysts compared to other reported catalysts, as shown in Supplementary Table 8. However, it is important to note that comparing these results with those from different test conditions may not be entirely fair. Furthermore, the site density and utilization rate of the catalysts are notably higher compared to the materials listed in Table 8. Could you please provide an explanation for this observation?

Reviewer #2 (Remarks to the Author):

The authors have addressed my concerns and the current manuscript is in a publishable form.

Reviewer #3 (Remarks to the Author):

The manuscript has been well revised and can be accepted for publication.

Manuscript NCOMMS-23-53238B-point-by-point responses to the reviewer's comments

Reviewer #1 (Remarks to the Author):

Authors well explained about the reviewer's comments and make up the additional analysis data for mechanistic study. The originality of this paper lies in systematically confirming the performance changes of the catalyst due to electron-withdrawing/donating effects and demonstrating reaction mechanisms and catalyst characteristics through various analyses. Although, we have minor concern, this paper could be considered for publication in *Nature Communications*.

Response: We appreciate your summary of the manuscript and the encouraging comment that our manuscript could be accepted for publication in *Nature Communications* after minor revision. The answers to the questions you raised are addressed in detail as below.

1. The authors demonstrated a high site density and turnover frequency of the synthesized catalysts compared to other reported catalysts, as shown in Supplementary Table 8. However, it is important to note that comparing these results with those from different test conditions may not be entirely fair. Furthermore, the site density and utilization rate of the catalysts are notably higher compared to the materials listed in Table 8. Could you please provide an explanation for this observation?

Response: Thank you for this important suggestion. The reviewer mentioned that the site densities of the catalysts are higher compared to the materials listed in comparison summary table (Supplementary Table 3). It is widely recognized that multiple technologies have been developed to evaluate the physically meaningful site density and turnover frequency (TOF) values from the experimental perspective, including CO adsorption, nitrite reduction methods. The obtained value of site density by different methods may vary significantly, since the estimation on site density depends on the interaction of the site-specific adsorbates with the surface of catalytic center, and the variation in electrochemical potential and experimental conditions will influence the adsorption/desorption events. Generally, CO adsorption overestimated site density while nitrite reduction underestimated it (please refer to *Nature Catalysis* **2022**, 5, 163–170, *JACS Au* **2021**, 1, 586-597).

Differently, to monitor the dynamic behavior and assess the value of active sites that **truly participate in the electrocatalytic reaction**, the scanning electrochemical microscopy (SECM) technology has been developed (please refer to *Nature Catalysis* **2021**, 4, 615–622; *ACS Energy Lett.* **2019**, 4, 1793–1802; *Angew. Chem. Int. Ed.* **2020**, 59, 16376–16380; *Acc. Chem. Res.* **2022**, 55, 5, 759–769). Considering that, in our work, the SECM technique was utilized to provide a straightforward approach for quantifying the **electrochemically accessible active sites that truly participate in ORR** with high resolution. Therefore, we believe that our site density and TOF values obtained by SECM technology are scientific. Also, it is indeed important to emphasize the test conditions for better comparing the site density results, thereby we revised and

added the corresponding measurement methods in Supplementary Table 3.

Supplementary Table 3. Comparison of SD_{mass} and TOF for CoCOP-COOH@KB, CoCOP-CH₃@KB, CoCOP-H@KB catalyst and reported catalysts in literatures.

Sample	SD (site g ⁻¹)	TOF (e site ⁻¹ s ⁻¹)	Measurement method	Ref.
CoCOP-COOH@KB	7.27×10 ¹⁹	4.60 @ 0.8 V	SECM	This work
CoCOP-CH ₃ @KB	2.38×10 ¹⁹	0.60 @ 0.8 V	SECM	This work
CoCOP-H@KB	3.36×10 ¹⁹	0.70 @ 0.8 V	SECM	This work
CoTAA-Ph(Cl)@GR	5.03×10 ¹⁹	0.45 @ 0.8 V	SECM	2
PANI-Co	4.22×10 ¹⁹	0.01 @ 0.8 V	CO adsorption	3
ZIF-Co	4.14×10 ¹⁹	0.11 @ 0.8 V	CO adsorption	3
FeN ₄	4.90×10 ¹⁹	0.04 @ 0.8 V	Neutron activation analysis	4
CFeN ₂	1.00×10 ¹⁹	0.01 @ 0.8 V	Neutron activation analysis	4
Fe-NC ^Δ -DCDA	4.69×10 ¹⁹	0.13 @ 0.85 V	Nitrite reduction	5
Fe _{0.5} NC-800	3.99×10 ¹⁹	0.46 @ 0.9 V	CO adsorption	6
PAJ	2.00×10 ¹⁹	0.7 @ 0.8 V	Nitrite reduction	7
CNRS	6.00×10 ¹⁹	0.2 @ 0.8 V	Nitrite reduction	7

For calculating the TOF values in Supplementary Table 3, the TOF and kinetic mass activity (j_m) values were calculated by the following equations:

$$\text{TOF} = \frac{N_A \times j_m}{SD_{\text{mass}} \times F}$$

$$j_m = \frac{j_k}{m_{\text{catalyst}}}$$

where F represents the Faraday's constant (96485 C mol⁻¹), N_A is the Avogadro constant (6.02 × 10²³ mol⁻¹), SD_{mass} is the number of sites per catalyst mass, m_{catalyst} is the catalyst loading on the glassy carbon disc (mg cm⁻²) and j_k is kinetic current density.

For other catalysts listed in the Supplementary Table 3, they all exhibit a similar or lower $E_{1/2}$ to CoCOP-COOH@KB, such as FeNC-800 with the $E_{1/2}$ of 0.88V, however the lower TOF values of FeNC-800 was largely attributed to its sampling under the

higher potential (0.9 V). Also, we should emphasize that, as shown in the above equations, the TOF value is directly proportional to kinetic current density (j_k , which is dependent on the applied bias voltages). Generally, when j_k at a more negative potential is chosen for calculating TOF, the corresponding TOF value will sharply increase. In our work, we performed the TOF value calculation based on the potential of 0.8 V according to most of the reported literatures. Hence, this is the main reason that the CoCOP-COOH@KB claims a higher TOF for Co active sites than other catalysts in literatures.

Reviewer #2 (Remarks to the Author):

The authors have addressed my concerns and the current manuscript is in a publishable form.

Response: We sincerely thank for your professional comments to improve the quality of our manuscript and the recommendation that our manuscript could be accepted for publication in *Nature Communications*.

Reviewer #3 (Remarks to the Author):

The manuscript has been well revised and can be accepted for publication.

Response: We appreciate your valuable suggestions on our manuscript and the recommendation that our manuscript could be accepted for publication in *Nature Communications*.